# State Entropy Regularization for Robust Reinforcement Learning

**Yonatan Ashlag**[*]
Technion

**Uri Koren**[*]
Technion

**Mirco Mutti**
Technion

**Esther Derman**
MILA Institute

**Pierre-Luc Bacon**
MILA Institute

**Shie Mannor**
Technion, NVIDIA Research

## Abstract

State entropy regularization has empirically shown better exploration and sample complexity in reinforcement learning (RL). However, its theoretical guarantees have not been studied. In this paper, we show that state entropy regularization improves robustness to structured and spatially correlated perturbations. These types of variation are common in transfer learning but often overlooked by standard robust RL methods, which typically focus on small, uncorrelated changes. We provide a comprehensive characterization of these robustness properties, including formal guarantees under reward and transition uncertainty, as well as settings where the method performs poorly. Much of our analysis contrasts state entropy with the widely used policy entropy regularization, highlighting their different benefits. Finally, from a practical standpoint, we illustrate that compared with policy entropy, the robustness advantages of state entropy are more sensitive to the number of rollouts used for policy evaluation.

## 1 Introduction

Despite the impressive success of RL across various synthetic domains [29, 37, 51, 6], challenging issues still need to be addressed before RL methods can be deployed in the real world. Some of these challenges involve imperfect models, noisy observations, and limited data. In such settings, policies trained on *nominal* environments may perform poorly when faced with deviations from the assumed dynamics or reward structure at deployment [34]. This issue has led to a growing interest in *robust* RL, which aims to ensure reliable performance despite model misspecification or uncertainty [38, 46, 59, 31, 47]. Over the years, the well-established connection between regularization and robustness in machine learning has been extended to the RL setting [61, 19, 21, 11], where regularization has been shown to induce robustness to adversarial perturbations of the reward function [12, 5] and the transition kernel [10]. This duality has particularly been studied for policy entropy regularization [13].

Recent works [23, 50, 65, 25, 4] proposed to regularize the standard RL objective with the entropy of the distribution over states induced by a policy [18], either alone or in combination with policy entropy. This has empirically shown better exploration and consequently, improved sample efficiency [50]. However, to our knowledge, formal studies on the effect of state entropy regularization and its connection with robustness do not exist in the literature yet. Thus, an open question is whether robustness is a by-product of state entropy regularization just like it is for policy entropy, and if it is, what type of robustness results from state entropy regularization?

---

[*]Equal contribution. Correspondence to: yonatan.ashlag@gmail.com, uri.koren@campus.technion.ac.il

39th Conference on Neural Information Processing Systems (NeurIPS 2025).

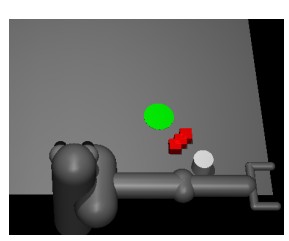
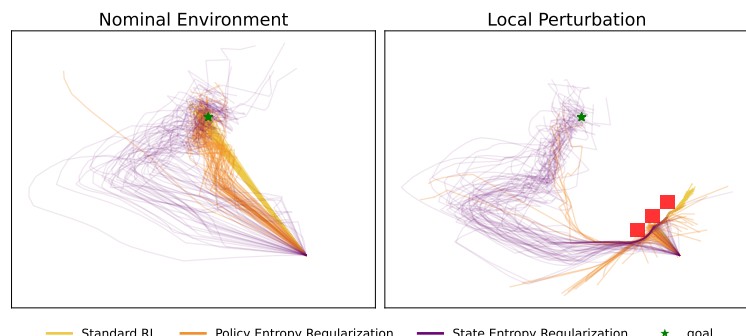

Figure 1: **State entropy regularization leads to diverse behavior and improved robustness.** We plot 50 trajectories from standard RL, policy-entropy–regularized RL and state-entropy–regularized RL on a manipulation task (left). Policies are trained on the nominal environment and evaluated in a perturbed version with a red obstacle along the optimal trajectory. The state-entropy–regularized agent often bypasses the obstacle and solves the perturbed task.

Intuitively, policy entropy encourages stochasticity in action selection but typically spreads randomness along a single dominant trajectory. This makes it effective in smoothing out small or uniform perturbations—but also fragile when that trajectory is disrupted. For instance, a large obstacle blocking a single high-reward path can cause policy entropy–regularized agents to fail catastrophically, as illustrated in Fig. 1. In contrast, state entropy incentivizes broader coverage of the state space, potentially distributing visitation across multiple high-reward paths. If policy entropy can be understood as a methodical way of injecting noise along an optimal path, state entropy can be viewed as encouraging uniformity across optimal, or nearly optimal, trajectories. This distinction motivates our investigation into whether state entropy regularization provably ensures stronger robustness, particularly in settings with spatially localized perturbations.

The contributions provided in the paper are summarized as follows.

- Section 3 provides a theoretical analysis of the *power* of state entropy regularization for robust RL:
  - In Section 3.1, we prove that state entropy regularization exactly solves a reward-robust RL problem, then characterize the induced uncertainty set and adversarial reward. Our theoretical result confirms the informal intuition described above: policy entropy protects against a locally informed adversary, while state entropy is robust to globally informed perturbations. We further show that the regularization strength monotonically controls the conservativeness of the induced uncertainty set, interpolating between $\ell_\infty$-like uncertainty in the low-regularization limit and $\ell_1$-like uncertainty as regularization increases.
  - In Section 3.2, we show that state entropy regularization provides a non-trivial lower bound on performance under transition (kernel) uncertainty. We compare this to the combined use of state and policy entropy, and prove that adding policy entropy weakens the bound—highlighting a structural advantage of using state entropy alone in this setting.
- In Section 4, we study both theoretical and practical *limitations* of entropy regularization:
  - We establish that entropy regularization—whether over the policy, the state distribution, or both—cannot solve any kernel robust RL problem (Section 4.1). Building upon this result, we show that entropy regularization can hurt risk-averse performance arbitrarily (Section 4.2);
  - We show that the robustness benefits of state entropy regularization can be more sensitive to the number of rollouts used to evaluate the policy w.r.t. other forms of regularization (Section 4.3).
- In Section 5, we empirically evaluate the robustness properties of state entropy regularization across discrete and continuous control tasks. We show that it improves performance under spatially correlated perturbations—specifically obstacle placement— while not degrading performance under smaller, more uniform perturbations. We also demonstrate how the robustness benefits of state entropy depend on the rollout budget, with diminished gains in low-sample regimes.

Related works are provided in Section 6. All proofs are provided in Appendix A.

## 2 Preliminaries

In this section, we describe the setting and notations we will use in the remainder of the paper.

**Notation.** We denote by $\Delta_{\mathcal{Z}}$ the set of probability distributions over a finite set $\mathcal{Z}$. The entropy of a distribution $\boldsymbol{\mu} \in \Delta_{\mathcal{Z}}$ is given by $\mathcal{H}_{\mathcal{Z}}(\boldsymbol{\mu}) := -\sum_{z \in \mathcal{Z}} \boldsymbol{\mu}(z) \log(\boldsymbol{\mu}(z))$. The log-sum-exp operator over $\mathcal{Z}$ is defined as $\mathrm{LSE}_{\mathcal{Z}}(\boldsymbol{u}) := \log\left(\sum_{z \in \mathcal{Z}} e^{\boldsymbol{u}(z)}\right)$, $\boldsymbol{u} \in \mathbb{R}^{\mathcal{Z}}$. For any $\boldsymbol{u}, \boldsymbol{v} \in \mathbb{R}^{\mathcal{Z}}$, their inner product is denoted by $\langle \boldsymbol{u}, \boldsymbol{v} \rangle_{\mathcal{Z}} := \sum_{z \in \mathcal{Z}} \boldsymbol{u}(z)\boldsymbol{v}(z)$. We shall often omit the set subscript if there is no ambiguity. Finally, for a distribution $\boldsymbol{\mu} \in \Delta_{\mathcal{Z}}$ and a function $f : \mathcal{Z} \to \mathbb{R}$, we denote the expectation of $f$ on $\boldsymbol{\mu}$ as $\mathbb{E}_{\boldsymbol{\mu}}[f(z)] = \sum_{z \in \mathcal{Z}}[\boldsymbol{\mu}(z)f(z)]$.

**Standard RL.** Markov Decision Processes (MDPs) constitute the formal framework for RL [54]. An MDP is a tuple $(\mathcal{S}, \mathcal{A}, P, r, \gamma)$, where $\mathcal{S}$ is a state space of size $S$, $\mathcal{A}$ an action space of size $A$, $P : \mathcal{S} \times \mathcal{A} \to \Delta(\mathcal{S})$ a transition kernel, $r : \mathcal{S} \times \mathcal{A} \to \mathbb{R}$ a reward function, and $\gamma \in [0, 1)$ a discount factor. The MDP evolves as follows. First, an initial state is sampled as $s_0 \sim P(\cdot)$.[2] Then, for every $t \geq 0$, the current state $s_t$ is observed, an action $a_t \sim \pi(\cdot|s_t)$ is performed according to some policy $\pi : \mathcal{S} \to \Delta_{\mathcal{A}}$, prompting a state transition $s_{t+1} \sim P(s_t, a_t)$ while a reward $r_t = r(s_t, a_t)$ is collected. The goal is to find a policy maximizing a specific objective function within a policy space $\Pi$. As we shall see below, different objectives can be considered.

First, define *expected* transition kernel and reward under policy $\pi$ as $P^{\pi}(\cdot|s) := \sum_{a \in \mathcal{A}} \pi(a|s)P(\cdot|s, a)$ and $r^{\pi}(s) := \sum_{a \in \mathcal{A}} \pi(a|s)r(s, a)$, sometimes denoted as $P^{\pi}, r^{\pi}$ in vector form. Also define the discounted *state distribution* induced by a policy $\pi$ on an MDP with kernel $P$ as $d_P^{\pi}(s) := \sum_{t=0}^{\infty} \gamma^t \mathbb{P}(s_t = s | \pi, P)$ and the state-action distribution, a.k.a. *occupancy* as $\rho_P^{\pi}(s, a) := d_P^{\pi}(s)\pi(a|s)$. The standard RL objective function to maximize over $\Pi$ is $\mathcal{J}(\pi, P, r) = \mathbb{E}_{\rho_P^{\pi}}[r(s, a)]$, which may be rewritten as $\mathcal{J}(\pi, P, r) = \langle \rho_P^{\pi}, r \rangle$ in vector form.

We note that, while the above formulation assumes finite MDPs, the main results extend to continuous state or action spaces under mild regularity assumptions. In that case, sums over states and actions are replaced by integrals with respect to the corresponding densities, and the discrete entropy is replaced by the differential entropy of the occupancy measure.

**Robust RL.** The robust RL formulation [59, 31] aims to tackle uncertainty in the MDP parameters, such that the kernel $P$ and reward $r$ are not known but assumed to lie in an uncertainty set $\mathcal{P} \times \mathcal{R}$. The objective function is defined with the worst case on the uncertainty set

$$\mathcal{J}(\pi, \mathcal{P}, \mathcal{R}) := \min_{(P, r) \in \mathcal{P} \times \mathcal{R}} \mathcal{J}(\pi, P, r).$$

By construction, this so-called robust MDP formulation leads to policies that are less sensitive to model misspecification and perform reliably under a range of plausible environments. Throughout this paper, we will often consider reward uncertainty sets $\{P\} \times \mathcal{R}$ of the form

$$\mathcal{R} = \{\tilde{r} \not\geq r \mid D(r, \tilde{r}) \leq \epsilon\}, \tag{1}$$

where $\epsilon > 0$, $D$ is interpreted as a dissimilarity measure between reward functions defining the notion of proximity within the uncertainty set and $\tilde{r} \not\geq r$ means that there exists some $(s, a) \in \mathcal{S} \times \mathcal{A}$ such that $\tilde{r}(s, a) \leq r(s, a)$ (to ensure the adversary can degrade performance rather than only improve it). Another perspective on robust RL is to interpret the minimization in $\mathcal{J}(\pi, \mathcal{P}, \mathcal{R})$ as the action of an *adversary* that perturbs the nominal MDP within the uncertainty set so as to minimize the policy's expected return. Accordingly, $\epsilon$ quantifies the adversary's *budget* for allowable perturbations.

**Entropy-regularized RL.** A broad range of literature considers augmenting the standard RL objective with some regularization function. The most popular is *policy entropy* regularization [17], which has been linked to stability of the policy optimization [2], improved exploration [45], and robustness [13]. The resulting objective is

$$\mathcal{J}_{\alpha}(\pi, \mathcal{H}_{\mathcal{A}}) := \mathbb{E}_{\rho_P^{\pi}}\left[r(s, a) + \alpha \mathcal{H}_{\mathcal{A}}(\pi(\cdot|s))\right]. \tag{2}$$

where $\alpha > 0$ is a hyperparameter controlling the regularization strength. In recent works [23, 50, 65, 25, 4], regularization with the entropy of the state distribution has been considered, either standalone or in combination with policy entropy, yielding

$$\mathcal{J}_{\alpha}(\pi, \mathcal{H}_{\mathcal{S}}) = \mathbb{E}_{\rho_P^{\pi}}[r(s, a)] + \alpha \mathcal{H}_{\mathcal{S}}(d_P^{\pi}) \tag{3}$$

$$\mathcal{J}_{\alpha}(\pi, \mathcal{H}_{\mathcal{S} \times \mathcal{A}}) = \mathbb{E}_{\rho_P^{\pi}}[r(s, a)] + \alpha \mathcal{H}_{\mathcal{S} \times \mathcal{A}}(\rho_P^{\pi}). \tag{4}$$

In this work, we will focus on analyzing the latter types of regularization.

---

[2]With a slight overload of notation, we denote the initial state distribution with the same symbol of the transition kernel with a void input.

| Regularizer | Uncertainty Set | Robust Return | Worst Case Reward |
|---|---|---|---|
| Policy | $\mathbb{E}_{d_P^\pi}\mathrm{LSE}_{\mathcal{A}}\left(\frac{\Delta r_s}{\alpha}\right) \leq \frac{\epsilon}{\alpha} + \log(A)$ | $\mathbb{E}_{\rho_P^\pi}[r] - \epsilon + \alpha\left(\mathbb{E}_{d_P^\pi}[\mathcal{H}_{\mathcal{A}}(\pi_s)] - \log(A)\right)$ | $r(s,a) - \alpha\log(\pi(a\mid s)) - \epsilon$ |
| State | $\mathrm{LSE}_{\mathcal{S}}\left(\frac{\Delta r^\pi}{\alpha}\right) \leq \frac{\epsilon}{\alpha} + \log(S)$ | $\mathbb{E}_{\rho_P^\pi}[r] - \epsilon + \alpha\left(\mathcal{H}_{\mathcal{S}}(d_P^\pi) - \log(S)\right)$ | $r(s,a) - \alpha\frac{\log(d_P^\pi(s))}{A\pi(a\mid s)} - \epsilon$ |
| State-action | $\mathrm{LSE}_{\mathcal{S}\times\mathcal{A}}\left(\frac{\Delta r}{\alpha}\right) \leq \frac{\epsilon}{\alpha} + \log(SA)$ | $\mathbb{E}_{\rho_P^\pi}[r] - \epsilon + \alpha\left(\mathcal{H}_{\mathcal{S}\times\mathcal{A}}(\rho_P^\pi) - \log(SA)\right)$ | $r(s,a) - \alpha\log(\rho^\pi(s,a)) - \epsilon$ |

Table 1: Robustness guarantees for each regularization type including the form of the uncertainty set, the robust return, and the corresponding adversarial reward. We denote $\Delta r := r - \tilde{r}$ and $r_s = r(s,\cdot), \pi_s = \pi(\cdot \mid s)$ for brevity.

# 3 Robustness and State Entropy Regularization

In this section, we analyze the robustness properties of regularization with the state distribution entropy. First, we show that this regularization exactly solves certain reward-robust RL problems, and we compare its theoretical guarantees to those of policy entropy regularization. We then extend the analysis to transition (kernel) uncertainty, deriving non-trivial performance bounds in this setting.

## 3.1 Reward-Robustness

In this section, we show that state entropy regularization amounts to solving a robust RL problem with an explicit uncertainty set over rewards.

**Theorem 3.1.** *For any $\pi \in \Pi$, the following duality holds:*

$$\min_{\tilde{r}\in\tilde{\mathcal{R}}} \mathbb{E}_{\rho^\pi}[\tilde{r}] = \mathbb{E}_{\rho^\pi}[r] + \alpha(\mathcal{H}_{\mathcal{S}}(d_P^\pi) - \log(S)) - \epsilon, \tag{5}$$

*where the reward uncertainty set is $\tilde{\mathcal{R}}^\pi(\epsilon, \alpha) := \left\{\tilde{r} \mid \mathrm{LSE}_{\mathcal{S}}\left(\frac{r^\pi - \tilde{r}^\pi}{\alpha}\right) \leq \frac{\epsilon}{\alpha} + \log(S)\right\}$. Moreover, the worst-case reward chosen by the adversary is given by $\tilde{r}(s,a) = r(s,a) - \frac{1}{A}\frac{\log(d_P^\pi(s))}{\pi(a\mid s)} - \epsilon$.*

The proof is presented in Appx. A.1. It follows a similar path to [5, 13] and relies on a convex conjugate argument: The entropy regularization term can be interpreted as introducing a soft constraint, corresponding to an uncertainty set over the reward function. The uncertainty set can be written as $\left\{\tilde{r} \not\succ r \mid \alpha\mathrm{LSE}_{\mathcal{S}}\left(\frac{r^\pi - \tilde{r}^\pi}{\alpha}\right) - \alpha\log(S) \leq \epsilon\right\}$, highlighting a dissimilarity constraint between reward vectors as in Eq. (1).

We compare the obtained uncertainty sets and robustness guarantees with policy entropy and state-action entropy regularization in Table 1. The comparison extends previous formulations that neglect either the temperature parameter $\alpha$ or the adversary's budget $\epsilon$ [13, 5]. More insight into the proofs of these derivations is included in Appx. A.2. As shown in the adversarial reward column, policy entropy regularization depends on the policy $\pi(\cdot|s)$ *only*. Thus, it tackles adversaries with access to local information of each state. In contrast, state entropy regularization depends on the state occupancy distribution $d_P^\pi$, enabling robustness against adversaries that exploit global properties of the agent's behavior across the state space. Consequently, policies regularized via state entropy are inherently more resistant to spatially-structured perturbations, while policy entropy provides protection primarily against local, state-specific variations. This difference analytically confirms our initial intuition.

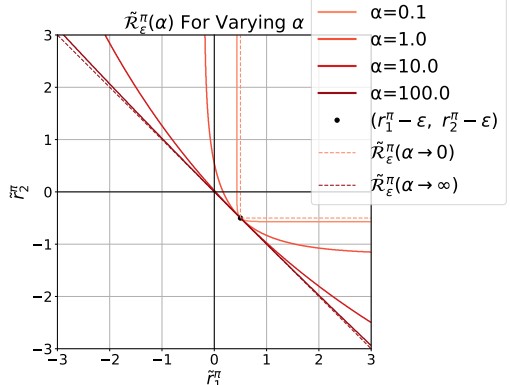

Figure 2: Uncertainty sets $\tilde{\mathcal{R}}_\epsilon^\pi(\alpha)$ induced by state entropy regularization in a two-state MDP with $r^\pi(s_1) = 1$, $r^\pi(s_2) = 0$, and $\epsilon = \frac{1}{2}$. The $x$- and $y$-axes correspond to the perturbed rewards $\tilde{r}^\pi(s_1)$ and $\tilde{r}^\pi(s_2)$, respectively. As $\alpha$ increases, the uncertainty set expands, interpolating between $\ell_\infty$- and $\ell_1$-type constraints in the limits $\alpha \to 0$ and $\alpha \to \infty$.

We next analyze how the uncertainty set induced by state entropy regularization evolves as a function of the temperature parameter $\alpha$. While our formal analysis focuses on the state entropy, similar

limiting behaviors appear to hold for policy entropy and state-action entropy as well. This formalizes insights that were observed empirically in [5], where varying the regularization strength was found to interpolate between different robustness behaviors. We characterize the limiting uncertainty sets as $\alpha \to \infty$ and $\alpha \to 0$, corresponding to different adversarial capabilities. The proof is in Appx. A.3. We visualize this behavior in Fig. 2.

**Theorem 3.2** (Limit uncertainty in $\alpha$). *Consider the uncertainty sets associated with state entropy regularization as a function of $\alpha$:*

$$\tilde{\mathcal{R}}_\epsilon^\pi(\alpha) = \left\{ \tilde{r} \mid \mathrm{LSE}_\mathcal{S}\left( \left( \frac{r^\pi - \tilde{r}^\pi}{\alpha} \right)^\alpha \right) - \alpha \log(S) \leq \epsilon \right\}.$$

*Then, the mapping $\alpha \mapsto \tilde{\mathcal{R}}_\epsilon^\pi(\alpha)$ is non-increasing in the sense that $\alpha_2 \geq \alpha_1$ implies $\tilde{\mathcal{R}}(\epsilon, \alpha_1) \subseteq \tilde{\mathcal{R}}(\epsilon, \alpha_2)$. Furthermore, the limit sets satisfy*

$$\lim_{\alpha \to \infty} \tilde{\mathcal{R}}_\epsilon^\pi(\alpha) = \cup_{\alpha > 0} \tilde{\mathcal{R}}_\epsilon^\pi(\alpha) = \left\{ \tilde{r} \,\middle|\, \frac{1}{S} \sum_{s \in \mathcal{S}} r^\pi(s) - \tilde{r}^\pi(s) \leq \epsilon \right\}$$

*and*

$$\lim_{\alpha \to 0} \tilde{\mathcal{R}}_\epsilon^\pi(\alpha) = \cap_{\alpha > 0} \tilde{\mathcal{R}}_\epsilon^\pi(\alpha) = \left\{ \tilde{r} \,\middle|\, \max_\mathcal{S}(r^\pi - \tilde{r}^\pi) \leq \epsilon \right\}.$$

## 3.2 Lower Bound under Kernel Uncertainty

As a first step in our analysis of kernel robustness, we show that state entropy regularization yields a non-trivial lower bound on worst-case performance under transition uncertainty.

**Theorem 3.3.** *For any policy $\pi \in \Pi$ and $\alpha > 0$, the following weak duality holds:*

$$\min_{\tilde{\mathcal{P}}^\pi(\epsilon)} \mathbb{E}_{\rho_{\tilde{P}}^\pi}[r] \geq \exp\left( \mathbb{E}_{\rho_P^\pi}[\log(r)] + \alpha(\mathcal{H}(d_P^\pi) - \log(S)) - \epsilon \right) \tag{6}$$

*where*

$$\tilde{\mathcal{P}}^\pi(\epsilon, \alpha) := \left\{ \tilde{P} \mid \alpha \log \left( \frac{1}{\mathcal{S}} \sum_{s \in \mathcal{S}} \left( \frac{d_P^\pi(s)}{d_{\tilde{P}}^\pi(s)} \right)^{\frac{1}{\alpha}} \right) \leq \epsilon \right\}$$

The proof follows a similar structure as in [13] where transition uncertainty is effectively reduced to reward uncertainty via importance sampling argument and logarithmic transformation. This reduction in turn enables us to leverage our earlier result on reward-robustness to derive the lower bound. The full proof is available in Appx. A.4.

An appealing property of this formulation is that the uncertainty set is expressed explicitly in terms of the induced state distribution. This allows us to reason about the distributional effects of transition noise directly, rather than working with abstract perturbations to the transition kernel itself.

Interestingly, using the same derivation, we can show that adding policy entropy regularization with the same temperature $\alpha$ leads to the same uncertainty set, but with a strictly looser lower bound. A more formal discussion of this comparison, including the explicit form of the resulting bound under policy entropy regularization, is provided in Appx. A.4.1. This result is particularly relevant in light of the next section, where we show that entropy regularization alone—regardless of the type—cannot solve kernel robust RL problems in general. In that sense, the bound derived here is the most we can expect from this approach. That said, we do not claim this lower bound is tight, and we cannot conclusively rule out that policy entropy may help in some settings. In fact, as we show in our experiments, it often does. Nonetheless, this result provides insight into the structural advantages of state entropy regularization when viewed through the lens of kernel uncertainty.

## 4 Limitations of State Entropy Regularization

In this section, we study the limitations of entropy regularization in robust reinforcement learning. First, we show that entropy regularization—whether over the policy, state distribution, or both—cannot solve any kernel robust RL problems. Building on this result, we demonstrate that in the risk-averse setting, entropy regularization can significantly degrade performance. Finally, we take a practical perspective and show that the robustness benefits of state entropy regularization are highly sensitive to the number of rollouts used during policy evaluation.

## 4.1 Entropy Regularization Cannot Fully Solve Kernel Robustness

A kernel-robust RL problem is defined by a function $\tilde{\mathcal{P}}(\text{MDP}, \pi)$ which maps the MDP and policy parameters to an uncertainty set—that is, the set of possible transition kernels from which the adversary may choose. We consider a class of kernel robust RL problems in which the uncertainty set function depends only on the nominal transition kernel $P$ and the policy $\pi$. This captures a wide range of uncertainty models, including $\ell_p$-constrained and rectangular uncertainty sets commonly studied in robust RL [27, 10]. We now state a more general claim: Any regularization function that depends only on the nominal kernel and the policy cannot, in general, characterize the robust return over such uncertainty set functions. The following impossibility result formalizes this observation.

**Theorem 4.1.** *Let $\Omega(\pi, P) \not\equiv 0$ be a regularization function. There does not exist an uncertainty set function $\tilde{\mathcal{P}} : (P, \pi)$ such that for every MDP holds:*

$$\min_{\tilde{P} \in \tilde{\mathcal{P}}} \mathbb{E}_{\rho_{\tilde{P}}^{\pi}}[r] = \mathbb{E}_{\rho_P^{\pi}}[r] + \Omega(\pi, P). \tag{7}$$

The proof, which can be found in Appx. A.5, is very simple and uses a symmetrization argument—when the reward is constant, any regularization must be degenerate. A key takeaway is that any regularization capable of capturing kernel robustness must depend on the reward, not just on the nominal kernel $P$ or the policy $\pi$. This observation is consistent with prior results [10]. As a direct corollary, entropy regularization—whether applied to the policy or the state distribution—cannot fully solve kernel robustness either.

## 4.2 Entropy Regularization and Risk-Averse Performance

While entropy regularization provides non-trivial lower bounds under kernel uncertainty, we have shown it cannot fully solve kernel robustness. Our experiments offer further insight into when regularization may improve robustness in practice. We now show analytically that for an important type of robustness—risk aversion—entropy regularization can, in fact, arbitrarily degrade performance.

While robust RL primarily addresses epistemic uncertainty—uncertainty about the parameters of the MDP—risk-averse RL [15] instead targets aleatoric uncertainty, which arises from the inherent randomness of the MDP itself. To quantify risk-averse performance, we use the common Conditional Value at Risk (CVaR) objective [48, 8], denoted $\text{CVaR}_{\beta}$, which measures the expected return in the worst $\beta$-fraction of outcomes. We denote the optimal policy for the unregularized objective by $\pi^*$ and the optimal policy for the entropy-regularized objective (whether state or policy entropy) by $\pi_R^*$.

**Theorem 4.2.** *For every temperature $\alpha > 0$, constant $M \in \mathbb{R}$, and $0 < \beta < 1$, there exists an MDP $\mathcal{M}$ such that:*

$$\text{CVaR}_{\beta}^{\pi^*}[G] - \text{CVaR}_{\beta}^{\pi_R^*}[G] = M \tag{8}$$

*where $G$ is the total discounted return.*

The intuition behind the result is that entropy regularization does not prioritize low-variance trajectories, which are critical for improving risk-averse performance. The proof is provided in Appx. A.6.

## 4.3 On the Sample Sensitivity of State Entropy Regularization

We now investigate how the robustification benefits of state entropy regularization depend on the number of rollouts used to estimate a policy's state entropy. While using state entropy regularization does not incur significant computational overhead [50], we observe that the number of rollouts required to realize the robustness benefits is higher than what is typically sufficient for standard RL. We provide both theoretical and empirical evidence for this phenomenon, and offer two complementary explanations. One explanation is statistical—state entropy is harder to estimate accurately than other quantities like return or policy entropy. The other is behavioral—in low-rollout regimes, entropy maximization can incentivize an agent to be less stochastic.

For the statistical explanation, we assume that the state entropy is estimated using a $k$-nearest neighbor estimator [52] in a continuous state space $\mathbb{R}^D$, as it is common in the related literature [42, 33, 50]. For fixed $k$, under mild regularity conditions on the MDP—such as the smoothness of $d_P^{\pi}$ for all the

policies $\pi$—the estimation error of the state entropy decays at a rate $O(n^{-1/D})$,[3] where $n$ denotes the number of sampled trajectories [53]. In contrast, both the policy entropy and expected return can be estimated with error decaying at the standard rate $O(n^{-1/2})$ [35]. As a result, in high-dimensional settings, state entropy estimation introduces a dominant source of statistical error, contributing to the increased sample sensitivity of entropy-regularized methods. In practice, we note that the effective dimensionality $D$ is often reduced by computing entropy over a subset of state features (as in Section 5) or by estimating entropy over latent state representations, as demonstrated by [50].

For the behavioral explanation, we consider the MDP illustrated in Figure 3 and the empirical objective $\mathbb{E}_{\rho^\pi}[r] + \alpha \hat{H}_1(\pi)$, where $\hat{H}_1(\pi)$ is the state entropy estimated on a single rollout. The policy that maximizes the empirical entropy on a single rollout will tend to be deterministic, taking the longest possible route to the goal in order to visit more states. This leads to a counterproductive outcome: Rather than encouraging stochasticity across multiple near-optimal paths—promoting robustness—the regularization yields a deterministic behavior along a suboptimal path, ultimately degrading robustness.

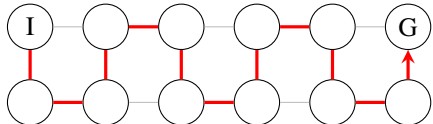

Figure 3: An illustrative toy MDP. The grid is a $2 \times 6$ environment with the initial state marked with "I" and the goal state with "G". The red path shows the longest deterministic trajectory that visits the maximum number of states before reaching the goal. Reward is given only at the goal state.

## 5  Experiments

We evaluate the robustness properties of state entropy regularization using a set of representative environments and perturbation types. These are designed to test three core hypotheses: (1) State entropy regularization offers protection against spatially correlated severe perturbations to the dynamics; (2) State entropy regularization induces robustness to $\ell_1$ reward uncertainty; (3) The robustification of state entropy regularization is sensitive to the number of rollouts used during training. We validate these claims in both discrete (MiniGrid [7]) and continuous (Mujoco [57]) domains. We compare policies trained without regularization, with policy entropy regularization, and with state entropy regularization. Note that the state-entropy regularized policies are trained with some policy entropy regularization to favor training stability [2].[4] For all the types, we select the largest regularization coefficients that degrade nominal performance less than 5%. To ensure a fair comparison, the methods are trained with the same base algorithm—A2C [36] for MiniGrid and PPO [49] for Mujoco.

To implement state entropy regularization, we largely follow [50] by using a k-nearest neighbor (k-NN) entropy estimator [52]. As state entropy regularization is incorporated as an intrinsic reward, it can be paired with any RL algorithm. Full implementation details are in Appx. B.1.

**Transition kernel robustness via entropy regularization.** To evaluate how well each regularization method handles spatially correlated perturbations to the transition dynamics, we construct environments where specific regions of the state space are blocked off or obstructed. These changes are considered "catastrophic" because they remove critical paths or create bottlenecks that fundamentally alter the agent's ability to reach the goal using its original policy. In MiniGrid, we randomly place a large wall segment that blocks a connected area of the maze (see Figure 4(b) for the perturbed environment). This constitutes a major spatially-structured disturbance, as it affects transitions between nearby states and introduces a sharp structural change in the environment. As shown in Figure 4(b), the agent trained with state entropy regularization significantly outperforms both policy entropy and unregularized baselines. We note that we report results for all horizon lengths to avoid misleading conclusions based on a single fixed task length. In cases where one method clearly dominates across the full horizon range, we can draw decisive conclusions. When the curves intersect or remain close, it indicates that the relative performance is more sensitive to task structure. To verify that this effect extends to continuous control, we run analogous experiments in Mujoco's Pusher and Ant environments by placing obstacles along the optimal trajectories, fully blocking them. Again, as shown in Figs 5(a), 6(c) state entropy regularization yields substantially more robust performance as its induced behavioral diversity enables bypassing these local catastrophes.

---

[3]For simplicity, here we consider the i.i.d. case in which the entropy estimate is computed with a single data point coming from each trajectory, sampled with a geometric distribution with parameter $\gamma$.

[4]A discussion of the interplay between state entropy and policy entropy regularization is in Appx. B.5.

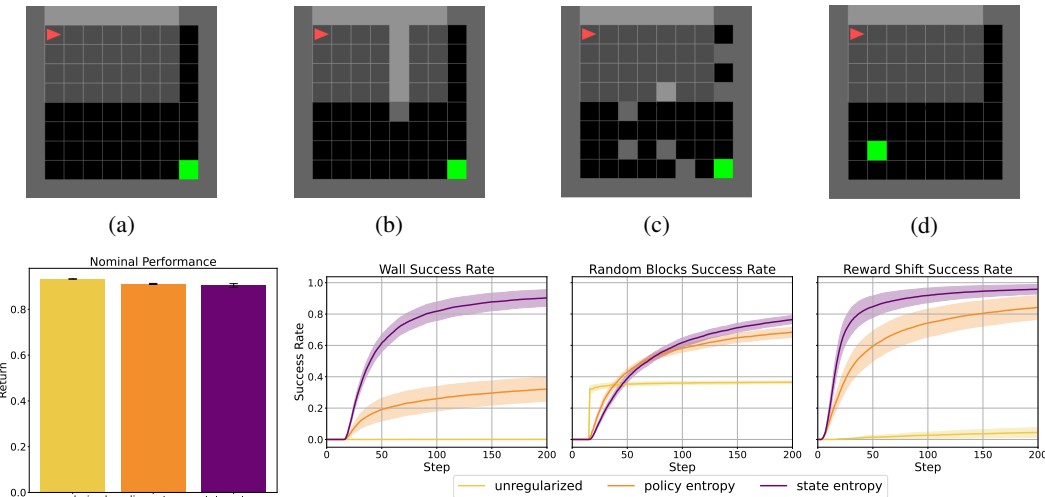

Figure 4: MiniGrid evaluation under different perturbations. All agents were trained on an empty grid where the agent starts at the top left and must reach the goal at the bottom right (a). Top: Visualizations of the three perturbation types used at test time—(b) large walls randomly block parts of the maze, (c) obstacles are distributed uniformly, and (d) the goal location is randomized. Bottom: Nominal return (left) and success rate under perturbations (right). We evaluate each policy for 200 episodes and report 96% confidence bounds across 25 seeds.

To highlight the challenge of learning policies that are robust to spatially localized perturbations, we compare against the PRMDP and NRMDP frameworks [6], two popular robustness baselines in continuous control. Both methods formulate robustness as a two-player zero-sum game: the agent selects actions while an adversary perturbs the actions. As shown in Figure 5(a), the agent trained with state entropy regularization significantly outperforms this baseline. While we do not expect state entropy to consistently outperform robust RL methods across all types of perturbations, this result—along with our analysis in the main paper—suggests that it is particularly effective against spatially correlated perturbations, which are not the primary focus of most standard approaches.

To test robustness under more uniformly distributed, small-scale perturbations, we augment MiniGrid with uniformly scattered obstacle tiles. These mild disturbances are spread across the entire state space, simulating the kind of noise that policy entropy regularization is expected to help with [13]. In Figure 4(c), we see that the unregularized agent performs poorly, while both policy and state entropy regularization improve robustness, with policy entropy showing a slight advantage. This result aligns with our initial intuition: Policy entropy is particularly effective against diffuse perturbations, and importantly, we observe that adding state entropy regularization does not hurt this effect.

**Reward robustness.** To assess how different entropy regularization methods handle reward uncertainty, we randomize the reward location at evaluation time in both MiniGrid and Pusher. In MiniGrid, a new goal location is sampled uniformly from the entire grid. In Pusher, the goal is relocated to a random position at distance $r$ from its nominal location. This setup induces structured variation in the reward function, formally resembling an $\ell_1$-bounded uncertainty—where the total change in reward mass is limited but can be concentrated arbitrarily across the state space. The key intuition is that if the reward happens to lie along the agent's typical trajectory, then policy entropy regularization—by injecting stochasticity at the action level—may suffice to recover it. However, when the reward is placed in less frequently visited areas, broader state coverage becomes necessary. In such cases, state entropy regularization offers a natural advantage by encouraging the agent to visit a wider range of states. As shown in Figures 4(d), 5 (b) policy entropy improves performance relative to the unregularized baseline, but adding state entropy yields a more substantial gain, particularly when the reward is far from the nominal goal location.

**Sensitivity to rollout budget.** We now examine how the number of rollouts used per state entropy estimation affects the robustness benefits of state entropy regularization. As discussed in Section 4.3, the estimation of state entropy is statistically more demanding than that of policy entropy or expected return. In particular, with limited rollouts, entropy maximization can encourage overly deterministic behavior, especially in sparse reward settings. To study this, we vary the number of

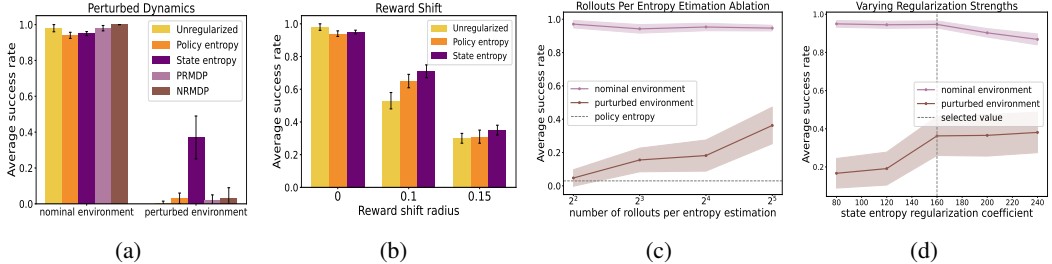

(a)  (b)  (c)  (d)

Figure 5: Pusher experiments and ablations. Like in the discrete experiment, we train all agents on a nominal task, then introduce a localized catastrophe by blocking off the optimal path, visualized in Fig. 1. All plots report success rate defined as when the pucks final distance from the goal is smaller than 0.1. We evaluate each policy for 200 episodes and show 96% confidence bounds across 25 seeds.

rollouts used to evaluate the policy and monitor how this affects downstream robustness. As shown in Figure 5(b), we find that when only a small number of rollouts are used, state entropy regularization can actually lead to worse performance than policy entropy. This is supported by the fact that with 4 rollouts, the confidence bounds of state entropy intersect the mean robust performance of policy entropy. Interestingly, while robustness is sensitive to the number of rollouts, nominal performance remains largely unchanged—making it difficult to detect this gap without explicitly evaluating under perturbations. These observations suggests that while state entropy can provide significant robustness benefits, realizing those benefits requires a sufficient amount of rollouts—underscoring a practical tradeoff between regularization and sample complexity.

**Regularization temperature.** To study how the temperature parameter influences robustness, we train state-entropy regularized agents with increasing regularization strength and evaluate their performance in both the nominal and perturbed versions of the Pusher environment. We observe in Figure 5(c) that as regularization increases, robust performance improves, illustrating the role of temperature in expanding the effective uncertainty set and promoting diverse robust behavior. Notably, robust performance eventually plateaus, suggesting diminishing returns from further increasing the regularization strength. This empirical trend aligns with our theoretical result in Theorem 3.2, which characterizes the limiting behavior of the uncertainty set. These findings highlight that while tuning entropy regularization can enhance robustness, its benefits saturate beyond a certain point.

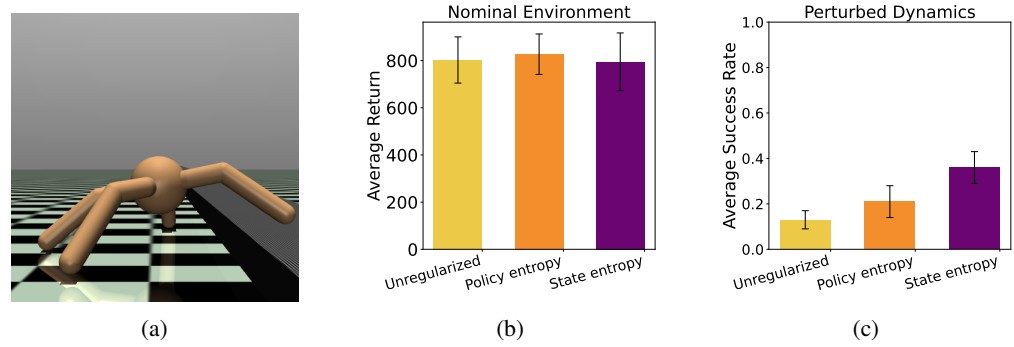

(a)  (b)  (c)

Figure 6: Ant experiment. (a) Visualization of the vertical wall perturbation. (b) All methods perform similarly on the nominal task. (c) Success rate in the perturbed environment, i.e., the fraction of evaluation runs in which the ant successfully crosses the obstacle. We evaluate each policy for 200 episodes and show 96% confidence bounds across 25 seeds.

# 6 Related Works

**Robust RL and regularization.** The connection between regularization and robustness is well-established in RL. The work [13] demonstrates an equivalence between policy-entropy regularized objectives and reward robustness and a lower bound for kernel robustness. We extend their analysis to state entropy regularization, providing more direct and general proofs, which also accounts for the regularization coefficient $\alpha$ (for both policy and state entropy). State-action entropy regularization is

analyzed by [5, 22], which as we show in Appx. A.4.1, equals the sum of state entropy plus policy entropy. Note that state-action entropy encodes the joint information provided by state entropy and policy entropy, but does not encode the conditional information each one gives to the other. Neither of these works provide algorithmic methods to implement state-action entropy regularization, nor studies the kernel robustness induced by entropy regularization. Complementary to ours, [10] derives a regularizer that depends on the value function since they study the primal optimization of RL and robust RL problems. However, a connection between regularized dual and primal objectives has yet to be established. The motivation in [10, 14] is also different from ours, as they study tractable robust RL methods, whereas we analyze robustness granted by state entropy regularization.

**State entropy objectives.** Hazan et al. [18] have first proposed a learning objective predicated on maximizing the entropy of the state distribution in MDPs, followed by a considerable stream of works studying various aspects of the problem, including practical algorithms for challenging domains [42, 33, 50, 64, 32], alternative formulations [43, 69, 16, 40, 65, 1], statistical complexity [56], how to deal with partial observability [66, 67], and many others [28, 23, 55, 63, 41, 44, 62, 39, 24, 25, 70, 30, 4, 9, 68]. Most of the works focus on the entropy of the state distribution although the latter is often paired with standard policy entropy regularization in practical methods [33, 50, 64, 32]. While the state entropy has been often considered a standalone objective for pure exploration [e.g., 55], data collection [63], or policy pre-training (most of the others), some works have employed occupancy entropy as a regularization for RL [23, 50, 65, 25, 4]. A common challenge for all of the previous is that the feedback on the state entropy is not directly available to the agent and must be estimated from data, leading to scalability issues. A crucial advancement has been made by incorporating non-parametric entropy estimators [52] in practical algorithms [42, 33, 50, 64].

## 7 Conclusion

This work provides a comprehensive analysis of state entropy regularization through the lens of robustness. We show that it exactly solves certain reward-robust RL problems and induces stronger worst-case guarantees than policy entropy in structured perturbation settings. In the case of kernel uncertainty, we establish a non-trivial performance lower bound and highlight that adding policy entropy weakens this guarantee. At the same time, we prove fundamental limitations: entropy regularization cannot solve general kernel-robust problems and may degrade performance in risk-averse scenarios. On the practical side, we show that while state entropy improves robustness to spatially-structured changes, it does not harm performance under smaller uniform perturbations—though its effectiveness is sensitive to the number of policy evaluation rollouts. Together, our findings clarify when and how state entropy regularization can contribute to robust RL, and where its boundaries lie.

## Acknowledgment

The work of Yonatan Ashlag is supported by the Israel Science Foundation (ISF), grant No. 447/20. The work of Mirco Mutti is funded by the European Union (ERC, grant agreement No. 101041250).

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

# A Proofs

## A.1 Proof of Theorem 3.1

*Proof.* Since this proof does not involve adversarial perturbations to the transition kernel, we omit the subscript and write $d^\pi$ instead of $d_P^\pi$ for clarity. We will first show a result for the uncertainty set $\tilde{\mathcal{R}}(\epsilon, \pi) = \{\tilde{r} \mid \text{LSE}_{\mathcal{S}}(r^\pi - \tilde{r}^\pi) \leq \epsilon\}$, and then get the result from our main theorem using change of variables.

We fix $\pi$. First, we reformulate the optimization problem in terms of $\Delta r := r - \tilde{r}$:

$$\min_{\tilde{r} \in \tilde{R}} \mathbb{E}_{\rho^\pi}[\tilde{r}] = \mathbb{E}_{\rho^\pi}[r] + \min_{\Delta r \in \tilde{R}} -\mathbb{E}_{\rho^\pi}[\Delta r] \tag{9}$$

Note that $\tilde{\mathcal{R}}(\epsilon, \pi)$ is strictly feasible and that the objective and constraints are convex, therefore the optimal point satisfies the KKT conditions. Furthermore, the objective is linear in $\tilde{r}$, thus the minimum is achieved at the edge of the constraint with $\lambda > 0$. The Lagrangian is:

$$L(\Delta r, \lambda) = -\langle \rho^\pi, \Delta r \rangle + \lambda(\text{LSE}(\Delta r^\pi) - \epsilon) \tag{10}$$
$$\nabla_{\Delta r(s,a)} L = -\rho^\pi(s,a) + \lambda \pi(a \mid s)\text{softmax}(\Delta r^\pi)(s) = 0 \tag{11}$$
$$-d^\pi(s) + \lambda \text{softmax}(\Delta r^\pi)(s) = 0 \tag{12}$$

Where we get that last line by taking the sum over $a \in \mathcal{A}$ of both sides. By taking the sum over $s \in \mathcal{S}$ we can note that the last line implies that $\lambda^* = 1$ (softmax and $d^\pi$ are probability vectors). Therefore, our original optimization objective from 9 becomes:

$$\min_{\Delta r} -\langle \rho^\pi, \Delta r \rangle + \text{LSE}_{\mathcal{S}}(\Delta r^\pi) - \epsilon = -\epsilon + \min_{\Delta r^\pi} -\langle d^\pi, \Delta r^\pi \rangle + \text{LSE}_{\mathcal{S}}(\Delta r^\pi) \tag{13}$$

Finally, noting that entropy is the convex conjugate of LSE we get:

$$\min_{\tilde{r} \in \tilde{\mathcal{R}}} \mathbb{E}_{\rho^\pi}[\tilde{r}] = \mathbb{E}_{\rho^\pi}[r] - \epsilon + \min_{\Delta r^\pi} -\langle d^\pi, \Delta r^\pi \rangle + \text{LSE}_{\mathcal{S}}(\Delta r^\pi) = \mathbb{E}_{d^\pi}[r] + \mathcal{H}_{\mathcal{S}}(d^\pi) - \epsilon \tag{14}$$

We also note that substituting $\Delta r(s,a) = \frac{\log(d^\pi(s))}{\pi(a|s)A} + c$ for every constant $c$ satisfies equation 11 and is thus the adversarial reward. From the constraint $\text{LSE}(\Delta r^\pi) = \epsilon$ we can deduce that $c = \epsilon$.

Finally, we can now substitute $r \leftarrow \frac{r}{\alpha}, \tilde{r} \leftarrow \frac{\tilde{r}}{\alpha}, \epsilon \leftarrow \frac{\epsilon}{\alpha} + \log(S)$ and get the required result for different temperatures. □

## A.2 Proof of results in Table 1

*Proof.* We proved the result for state entropy in the previous appendix. For both policy entropy and state-action entropy we'll start from equation 9. Note that for both we can imply the KKT conditions since $\tilde{R}$ is strictly feasible and the objective and constraints are convex. For policy entropy the gradient of the Lagrangian becomes:

$$\nabla_{\Delta r(s,a)} L = -\rho^\pi(s,a) + \lambda d^\pi(s)\text{softmax}(\Delta r_s)(a) = 0 \tag{15}$$

By summing over $(s,a)$ we can again infer that $\lambda = 1$. We continue by substituting in the identity $\rho^\pi(s,a) = d^\pi(s)\pi(a \mid s)$ and we get $\pi(a \mid s) = \text{softmax}(\Delta r_s)(a)$. By taking the log of both sides and taking the inner product with $\rho^\pi$ we get:

$$-\langle \rho^\pi, \log(\pi(a \mid s)) \rangle = -\sum_s d^\pi(s) \sum_a \pi(a \mid s) \log(\pi(a \mid s))) = \mathbb{E}_{d^\pi}[\mathcal{H}_{\mathcal{A}}(\pi(\cdot \mid s))] \tag{16}$$

$$= \langle \rho^\pi, -\Delta r + \text{LSE}_{\mathcal{A}}(r_s) \rangle = \langle \rho^\pi, -\Delta r \rangle + \mathbb{E}_{d^\pi}[\text{LSE}_{\mathcal{A}}(\Delta r_s)] \tag{17}$$

Which is the required result for the case where $\alpha = 1$. We use the same change of variables as A.1 to get the result for a general temperature. We can verify with through explicit substitution that the worst case reward described in the table satisfies 15 and the requirement that $\mathbb{E}_{d^\pi}[\text{LSE}_{\mathcal{A}}(\Delta r_s)] = \epsilon$. The proof for state-action entropy regularization is very similar and is thus omitted. □

## A.3 Proof of Theorem 3.2

*Proof.* First, by taking the derivative of the divergence according to alpha we can see that it is always negative, which implies the required containment relation:

$$\frac{\partial}{\partial \alpha}\left(\alpha \log\left(\frac{1}{\mathcal{S}}\sum_s e^{\frac{\Delta r^\pi}{\alpha}}\right)\right) = \log\left(\frac{1}{\mathcal{S}}\sum_s e^{\frac{\Delta r^\pi}{\alpha}}\right) - \alpha\frac{\frac{1}{\mathcal{S}}\sum_s e^{\frac{\Delta r^\pi}{\alpha}}\frac{\Delta r^\pi}{\alpha^2}}{\frac{1}{\mathcal{S}}\sum_s e^{\frac{\Delta r^\pi}{\alpha}}} \tag{18}$$

$$= \log\left(\mathbb{E}_{s\sim U}[e^{\frac{\Delta r^\pi}{\alpha}}]\right) - \mathbb{E}_{s\sim\text{softmax}(\frac{\Delta r^\pi}{\alpha})}\left[\frac{\Delta r^\pi}{\alpha}\right] \tag{19}$$

$$\leq \log\left(\mathbb{E}_{s\sim U}[e^{\frac{\Delta r^\pi}{\alpha}}]\right) - \log\left(\mathbb{E}_{s\sim\text{softmax}(\frac{\Delta r^\pi}{\alpha})}\left[e^{\frac{\Delta r^\pi}{\alpha}}\right]\right) \tag{20}$$

$$\leq 0 \tag{21}$$

The first inequality follows from Jensen, the second inequality follows from proprieties of softmax (gives higher weight to higher $\Delta r^\pi$ values) and from the monotonicity of the exponent and log function.

Next, we prove the limiting case when $\alpha \to \infty$. Using L'Hôpital's rule we get:

$$\lim_{\alpha\to\infty}\alpha\log\left(\frac{1}{S}\sum_s e^{\frac{\Delta r^\pi}{\alpha}}\right) = \lim_{\alpha\to\infty}\frac{-\frac{\frac{1}{S}\sum_s e^{\frac{\Delta r^\pi}{\alpha}}\frac{\Delta r^\pi}{\alpha^2}}{\frac{1}{S}\sum_s e^{\frac{\Delta r^\pi}{\alpha}}}}{-\frac{1}{\alpha^2}} \tag{22}$$

$$= \lim_{\alpha\to\infty}\mathbb{E}_{s\sim\text{softmax}(\frac{\Delta r^\pi}{\alpha})}\left[\Delta r^\pi\right] \tag{23}$$

$$= \frac{1}{\mathcal{S}}\sum_s \Delta r^\pi(s) \tag{24}$$

Where the last equality follows from the fact that as $\alpha$ goes to infinity, the softmax term approaches the uniform distribution.

We finally prove the limiting case for $\alpha \to 0$. Again using L'Hôpital's rule:

$$\lim_{\alpha\to\infty}\alpha\log\left(\frac{1}{S}\sum_s e^{\frac{\Delta r^\pi}{\alpha}}\right) = \lim_{\alpha\to\infty}\mathbb{E}_{s\sim\text{softmax}(\frac{\Delta r^\pi}{\alpha})}\left[\Delta r^\pi\right] = \max \Delta r^\pi \tag{25}$$

$$\square$$

Where the final equality follows from the fact that as $\alpha$ goes to zero the softmax term is equivalent to taking a hard maximum over $\Delta r$.

## A.4 Proof of Theorem 3.3

*Proof.* First, we want to transform kernel uncertainty into reward uncertainty:

$$\log\mathbb{E}_{\rho_{\tilde{P}}^\pi}[r(s,a)] = \log\mathbb{E}_{\rho_P^\pi}\left[\frac{\rho_{\tilde{P}}^\pi}{\rho_P^\pi}r(s,a)\right] \qquad \text{(Importance sampling)} \tag{26}$$

$$\geq \mathbb{E}_{\rho_P^\pi}\left[\log(r(s,a))\right] + \mathbb{E}_{\rho_P^\pi}\left[\log\frac{\rho_{\tilde{P}}^\pi}{\rho_P^\pi}\right] \qquad \text{(Jensen)} \tag{27}$$

$$= \mathbb{E}_{\rho_P^\pi}\left[\log(r(s,a))\right] + \mathcal{H}_{\mathcal{S}\times\mathcal{A}}(\rho_P^\pi) + \mathbb{E}_{\rho_P^\pi}\left[\log\rho_{\tilde{P}}^\pi\right] \tag{28}$$

We think of $\tilde{r}' = \log\rho_{\tilde{P}}^\pi$ as adversarial reward and $r' = \log\rho_P^\pi$ as nominal reward. We thus have:

$$\Delta r' := r' - \tilde{r}' = \log\left(\frac{\rho_{\tilde{P}}^\pi}{\rho_P^\pi}\right) =: \log\left(\frac{d_{\tilde{P}}^\pi}{d_P^\pi}\right) \tag{29}$$

We further observe that:

$$\alpha\text{LSE}_{\mathcal{S}}\left(\frac{\Delta r^\pi}{\alpha}\right) - \alpha\log(\mathcal{S}) = \alpha\log\left(\frac{1}{\mathcal{S}}\sum_{s\in\mathcal{S}}\left(\frac{d_P^\pi(s)}{d_{\tilde{P}}^\pi(s)}\right)^{\frac{1}{\alpha}}\right) \leq \epsilon \tag{30}$$

Thus $\tilde{r}' \in \tilde{R}'^\pi(\epsilon, \alpha)$, where $\tilde{R}'^\pi(\epsilon, \alpha)$ is the reward uncertainty set described in Theorem 3.1. This containment relation implies:

$$\min_{\tilde{P} \in \tilde{\mathcal{P}}^\pi(\epsilon,\alpha)} \mathbb{E}_{\rho_{\tilde{P}}^\pi} \left[ \log \rho_{\tilde{P}}^\pi \right] \geq \min_{\tilde{r}' \in \tilde{R}'^\pi(\epsilon,\alpha)} \mathbb{E}[\tilde{r}'] = \mathbb{E}_{\rho_P^\pi}[\log(\rho_P^\pi)] + \alpha(\mathcal{H}_\mathcal{S}(d_P^\pi) - \log(\mathcal{S})) - \epsilon \quad (31)$$

$$= -\mathcal{H}_{\mathcal{S} \times \mathcal{A}}(\rho_P^\pi) + \alpha(\mathcal{H}_\mathcal{S}(d_P^\pi) - \log(\mathcal{S})) - \epsilon \quad (32)$$

Plugging this back in to 28 we get:

$$\min_{\tilde{P} \in \tilde{\mathcal{P}}^\pi(\epsilon,\alpha)} \log \mathbb{E}_{\rho_{\tilde{P}}^\pi}[r(s,a)] \geq \min_{\tilde{P} \in \tilde{\mathcal{P}}^\pi(\epsilon,\alpha)} \mathbb{E}_{\rho_P^\pi} \left[ \log(r(s,a)) \right] + \alpha(\mathcal{H}_\mathcal{S}(d_P^\pi) - \log(\mathcal{S})) - \epsilon \quad (33)$$

We finish the proof by taking the exponent of both sides.

$\square$

### A.4.1 Policy entropy lower bound

We aim to use the same methods as the in the proof of the above theorem, except this time for the regularization $\alpha\mathcal{H}_\mathcal{S}(d^\pi) + \alpha\mathbb{E}_{d^\pi}[\mathcal{H}_\mathcal{A}\pi(\cdot \mid s)]$. First, we note that from the chain rule we have $\mathcal{H}_{\mathcal{S} \times \mathcal{A}}(\rho^\pi) = \mathcal{H}_\mathcal{S}(d^\pi) + \mathbb{E}_{d^\pi}[\mathcal{H}_\mathcal{A}\pi(\cdot \mid s)]$. Therefore, we are actually using state-action entropy regularization with temperature $\alpha$.

We consider the same lower bound as in equation 28 and the same definition of $\Delta r'$ as in 29. The uncertainty set $\tilde{\mathcal{R}}^\pi(\epsilon, \alpha)$ corresponding to state-action entropy regularization is (see Table 1):

$$\alpha\text{LSE}_{\mathcal{S} \times \mathcal{A}} \left( \frac{\Delta r}{\alpha} \right) - \alpha\log(\mathcal{S}\mathcal{A}) = \alpha\log \left( \sum_{s,a \in \mathcal{S} \times \mathcal{A}} \left( \frac{d_P^\pi(s)}{d_{\tilde{P}}^\pi(s)} \right)^{\frac{1}{\alpha}} \right) - \alpha\log(\mathcal{S}\mathcal{A}) \quad (34)$$

$$= \alpha\log \left( \sum_{s \in \mathcal{S}} \left( \frac{d_P^\pi(s)}{d_{\tilde{P}}^\pi(s)} \right)^{\frac{1}{\alpha}} \right) - \alpha\log(\mathcal{S}) \leq \epsilon \quad (35)$$

Which is the exact same uncertainty set $\tilde{\mathcal{P}}^\pi(\epsilon, \alpha)$ as the one considered for state entropy alone. That being said, using the bounds in table 1 we get:

$$\min_{\tilde{P} \in \tilde{\mathcal{P}}^\pi(\epsilon,\alpha)} \log \mathbb{E}_{\rho_{\tilde{P}}^\pi}[r(s,a)] \geq \mathbb{E}_{\rho_P^\pi} \left[ \log(r(s,a)) \right] + \alpha(\mathcal{H}_{\mathcal{S} \times \mathcal{A}}(\rho_P^\pi) - \log(\mathcal{S}\mathcal{A})) - \epsilon \quad (36)$$

This is a worse lower bound than in the state entropy case, since:

$$\mathcal{H}_{\mathcal{S} \times \mathcal{A}}(\rho_P^\pi) - \log(\mathcal{S}\mathcal{A}) = (\mathcal{H}_\mathcal{S}(d_P^\pi) - \log(\mathcal{S})) + (\mathbb{E}_{d_P^\pi}[\mathcal{H}_\mathcal{A}(\pi(\cdot \mid s)] - \log(\mathcal{A})) \quad (37)$$

$$\leq \mathcal{H}_\mathcal{S}(d_P^\pi) - \log(\mathcal{S}) \quad (38)$$

### A.5 Proof of theorem 4.1

Let $\Omega \not\equiv 0$ a function. Assume towards contradiction that exists a function $\tilde{\mathcal{P}}(P)$ that satisfies $\min_{\tilde{P} \in \tilde{\mathcal{P}}} \mathbb{E}_{\rho_{\tilde{P}}^\pi}[r] = \mathbb{E}_{\rho_P^\pi}[r] + \Omega(\pi, P)$. We consider an MDP with reward vector $R \equiv c$ where $c \in \mathbb{R}$ is some constant. For all $\pi, P$ we have:

$$\min_{\tilde{P} \in \tilde{\mathcal{P}}} \mathbb{E}_{\rho_{\tilde{P}}^\pi}[r] = c = c + \Omega(\pi, P) \quad (39)$$

$$\Rightarrow \Omega(\pi, P) = 0 \quad (40)$$

But the last line implies $\Omega \equiv 0$, a contradiction.

### A.6 Proof of Theorem 4.2

Before the proof, we state the formal definition of the Conditional Value-at-Risk (CVaR). Let the return be defined as $R(\tau) := \sum_{t=0}^\infty \gamma^t r_t$ where the trajectory $\tau = (s_0, a_0, r_0, \dots)$ is sampled from

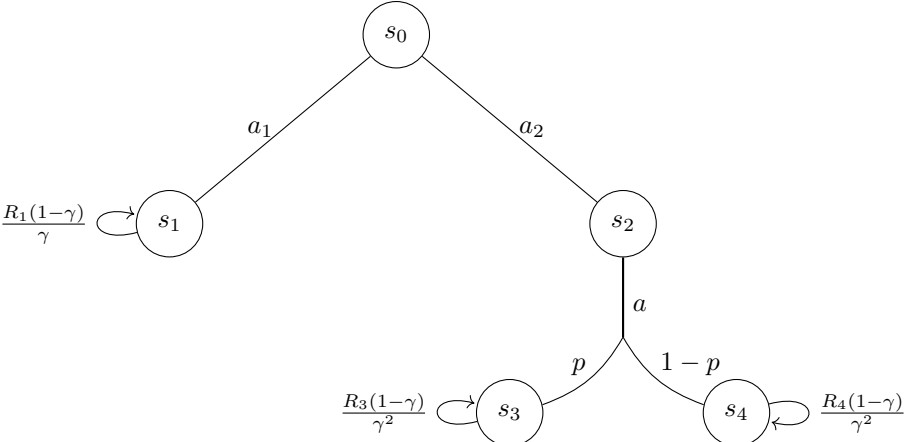

Figure 7: Counterexample used in the proof of Theorem 4.2. The initial state is $s_0$. All actions are deterministic except for action $a$ taken from state $s_2$, which leads stochastically to $s_3$ with probability $1 - p$ and to $s_4$ with probability $p$. Self-loops indicate absorbing states with associated returns.

$p_P^\pi(\tau) := \mathbb{P}[s_0, a_0, r_0, \ldots | \pi, P]$. Then, for a given confidence level $\beta \in (0, 1)$ the CVaR of the return under policy $\pi$ is defined as:

$$\mathrm{CVaR}_\beta^\pi[R(\tau)] := \min_{\eta \in \mathbb{R}} \left\{ \eta + \frac{1}{1 - \beta} \mathbb{E}_{p_P^\pi}[(R(\tau) - \eta)^+] \right\} \tag{41}$$

where $(x)^+ := \max(x, 0)$.

*Proof.* Fix some $M > 0$, $\alpha > 0$, and confidence level $0 < \beta < 1$. In this proof we'll denote the total discounted return by $G$. We construct an MDP (illustrated in Figure 7) that induces arbitrarily poor risk-sensitive performance under occupancy entropy regularization.

The key idea is to design an MDP in which two actions from the initial state, $a_1$ and $a_2$, lead to branches that are nearly identical in terms of expected return, but differ drastically in their return distributions. Specifically, we construct the branch associated with $a_2$ to be extremely risky, such that even a small degree of stochasticity—induced by entropy regularization—exposes the agent to severe downside risk.

More formally, we'll first require that:

$$\mathbb{E}[G \mid \pi(s_0) = a_1] - \epsilon = R_1 - \epsilon = \mathbb{E}[G \mid \pi(s_0) = a_2] = pR_3 + (1 - p)R_4 \tag{42}$$

for some $\epsilon > 0$, implying $\pi^*(a_1 \mid s_0) = 1$ and $\pi_R^*(a_1 \mid s_0) := q \approx \frac{1}{2}$ (the last claim is true because for every temperature $\alpha$ we can make $\epsilon$ arbitrarily small).

Second, we'll fix $R_1 = R > 0$, and since the branch associated with $a_1$ is deterministic we'll have $\mathrm{CVaR}_\beta^{\pi^*}[G] = \mathcal{J}(\pi) = R > 0$. We'll require that $\mathrm{CVaR}_\beta^{\pi_R^*}[G] \leq -M$ which clearly implies that the result from the proof holds.

We now show explicitly that exists $R_3, R_4, p$ s.t. the two conditions above hold. We focus on the second condition, and use the dual representation of CVaR [3]: $\mathrm{CVaR}_\beta^{\pi_R^*}[G] = \min_{Q \in \mathcal{U}} \mathbb{E}_{G \sim Q}[G]$ where $\mathcal{U} = \{Q \ll P \mid \forall G : \frac{Q(G)}{P(G)} \in (0, 1/\beta)\}$ and $P$ is the return distribution for $\pi_R^*$.

Since $1 > \beta > 0$, we can choose $1 > p > 0$ s.t. $(1 - q)p + q \geq \beta$. We construct a probability measure $Q \in \mathcal{U}$ s.t. $Q(R_3) = \frac{p(1-q)}{\beta}$ and $Q(R_1) = 1 - Q(R_3)$. Note that from the construction of $p$ we have that $Q(R_3) < 1$ and $1 - Q(R_3) = Q(R_1) \leq \frac{P(R_1)}{\beta} = \frac{q}{\beta}$ and indeed $Q \in \mathcal{U}$. We can now

analyze the risk averse performance of $\pi_R^*$:

$$\mathrm{CVaR}_\beta^{\pi_R^*}[G] \leq R_3 \frac{p(1-q)}{\beta} + R_1 \left( 1 - \frac{p(1-q)}{\beta} \right) \leq -M \tag{43}$$

$$R_3 \leq \frac{\beta(-M-R)}{p(1-q)} + R \tag{44}$$

We can choose an $R_3 = R'$ that satisfies the inequality above. Finally, we are only left with satisfying the first condition, and we can fix $R_4 = \frac{R-\epsilon-pR'}{1-p}$ and we are done. $\qquad\square$

### A.7 Robustness under estimation errors

In Section 3, we provided robustness guarantees for the reward uncertainty (Theorem 3.1) and kernel uncertainty (Theorem 3.3) of a policy maximizing the state-entropy regularized objective. In practice, however, the regularization term is not computed with the true state entropy but a sample-based estimate. It is then not obvious than similar guarantees hold under estimation error of the state entropy. In the following, we show how the guarantees generalize to the sample-based regime following a straightforward concentration argument on the state distribution. In the spirit of Section 3, we consider the entropy of a finite state space, while we refer to Section 4.3 for an informal discussion on the sample sensitivity in continuous settings, in which more sophisticated estimators are employed.

**Corollary A.1.** *Let $\hat{d}$ an empirical estimate of $d_P^\pi$ obtained with $n$ sampled trajectories, for which it holds $\|\hat{d} - d_P^\pi\|_1 \leq 1/2$. For $\delta \in (0,1)$ and any policy $\pi$, the following weak duality holds with probability $1 - \delta$*

$$\min_{\tilde{r} \in \tilde{\mathcal{R}}} \mathbb{E}_{\rho^\pi}[\tilde{r}] \geq \mathbb{E}_{\rho^\pi}[r] + \alpha \left( \mathcal{H}_\mathcal{S}(\hat{d}) - \sqrt{\frac{2S \log(2/\delta)}{n}} \log \sqrt{\frac{Sn}{\log(2/\delta)}} - \log(S) \right) - \epsilon,$$

*where the reward uncertainty set is $\tilde{\mathcal{R}}^\pi(\epsilon, \alpha) := \left\{ \tilde{r} \mid \mathrm{LSE}_\mathcal{S}(\frac{r^\pi - \tilde{r}^\pi}{\alpha}) \leq \frac{\epsilon}{\alpha} + \log(S) \right\}$.*

*Proof.* To prove the result, it is sufficient to show that the estimated entropy $\mathcal{H}_\mathcal{S}(\hat{d})$ concentrates around the true entropy $\mathcal{H}_\mathcal{S}(d_P^\pi)$. Let $\kappa := \|\hat{d} - d_P^\pi\|_1$, we write

$$|\mathcal{H}_\mathcal{S}(\hat{d}) - \mathcal{H}_\mathcal{S}| \leq \kappa \log S + \kappa \log \frac{1}{\kappa} + (1-\kappa) \log \frac{1}{1-\kappa} \tag{45}$$

$$\leq 2\kappa \log \frac{S}{\kappa} \tag{46}$$

$$\leq \sqrt{\frac{2S \log(2/\delta)}{n}} \log \sqrt{\frac{Sn}{\log(2/\delta)}} \qquad \text{[with probability } 1-\delta] \tag{47}$$

where the first inequality follows from continuity bounds of the entropy [60, see], the second inequality follows from the assumption $\kappa \leq 1/2$, the last inequality invokes concentration of the empirical distribution [58]. Finally, the result is straightforward by plugging equation 47 into the result of Theorem 3.1. $\qquad\square$

An analogous result for kernel uncertainty is as follows.

**Corollary A.2.** *Let $\hat{d}$ an empirical estimate of $d_P^\pi$ obtained with $n$ sampled trajectories, for which it holds $\|\hat{d} - d_P^\pi\|_1 \leq 1/2$. For any $\delta \in (0,1)$, policy $\pi$, and $\alpha > 0$, the following weak duality holds with probability $1 - \delta$*

$$\min_{\tilde{\mathcal{P}}^\pi(\epsilon)} \mathbb{E}_{\rho_{\tilde{P}}^\pi}[r] \geq \exp \left( \mathbb{E}_{\rho_P^\pi}[\log(r)] + \alpha \left( \mathcal{H}_\mathcal{S}(\hat{d}) - \sqrt{\frac{2S \log(2/\delta)}{n}} \log \sqrt{\frac{Sn}{\log(2/\delta)}} - \log(S) \right) - \epsilon \right),$$

*where*

$$\tilde{\mathcal{P}}^\pi(\epsilon, \alpha) := \left\{ \tilde{P} \mid \alpha \log \left( \frac{1}{\mathcal{S}} \sum_{s \in \mathcal{S}} \left( \frac{d_P^\pi(s)}{d_{\tilde{P}}^\pi(s)} \right)^{\frac{1}{\alpha}} \right) \leq \epsilon \right\}$$

*Proof.* The proof is straightforward by plugging equation 47 into the result of Theorem 3.3. $\qquad\square$

# B Experiments

The code for our experiments is available at `https://github.com/JonathanAshlag/State_entropy_robust_rl`.

## B.1 State entropy algorithm

**k-nearest neighbor entropy estimator.** Let $X$ be a random variable with a probability density function $p$ whose support is a set $\mathcal{X} \subset \mathbb{R}^q$ Then its differential entropy is given as $\mathcal{H}(X) = -\mathbb{E}_{x \sim p(x)}[\log(p(x))]$ When the distribution $p$ is not available, this quantity can be estimated given N i.i.d realizations of $\{x_i\}_{i=1}^N$. However, since it is difficult to estimate $p$ with high-dimensional data, particle-based k-nearest neighbors (k-NN) entropy estimator [52] can be employed:

$$\hat{H}_N^k(X) = \frac{1}{N} \sum_{i=1}^N \log \frac{N \cdot \|x_i - x_i^{k\text{-NN}}\|_2^{q/2} \cdot \hat{\pi}^{q/2}}{k \cdot \Gamma\left(\frac{q}{2} + 1\right)} + C_k \tag{1}$$

$$\hat{H}_N^k(X) \propto \frac{1}{N} \sum_{i=1}^N \log \|x_i - x_i^{k\text{-NN}}\|_2 \tag{2}$$

Where $x_i^{k\text{-NN}}$ is the k-NN of $x_i$ within a set $\{x_i\}_{i=1}^N$, $C_k = log(k) - \Psi(k)$ is a bias correction term, $\Psi$ the digamma function, $\Gamma$ the gamma function, q the dimension of x, $\hat{\pi} \approx 3.14159$, and the transition from (1) to (2) always holds for q > 0.

Using (2) the state entropy can be modeled into an intrinsic reward in the following way:

$$r^i(s_i) = \log(\|y_i - y_i^{k-nn}\| + 1) \tag{3}$$

where $y_i$ can be either $s_i$, a latent encoding of $s_i$ or selected features of $s_i$. The total reward is:

$$r_j^{total} = r(s_j, a_j) + \beta \cdot \gamma_t \cdot r^i(s_j) \tag{4}$$

Where $\beta$ is the regularization temperature, $\gamma_t$ is the intrinsic reward discount factor, it's usually the same as the extrinsic reward discount factor; we empirically find that in environments where the episode doesn't end upon task completion, a stronger intrinsic reward discount helps prevent over-exploration.

**Temperature warm-up.** We find that it helps performance to start training with a high temperature $\beta_{start}$ and anneal it through training. This is common in prior work [50], notably instead of exponential decay we use cosine decay from $\beta_{start}$ to $\beta > 0$

$$\beta_t = \beta + \frac{\beta_{\text{start}} - \beta}{2}\left(1 + \cos\left(\pi \tfrac{t-1}{T}\right)\right), \qquad t = 1, \ldots, T.$$

## B.2 Minigrid experiments

The experiment is built upon [26] open code base, which is released under the MIT License .[5]. Base algorithm is A2C. As the point of the expirement isnt handling partial observability, we remove potential bias by instead of using a random encoder, the features used for entropy estimation were the x,y position of the agent + one hot encoding of its direction. The nominal environment is an 8 by 8 empty grid with a sparse reward, 1 if goal is reached, 0 otherwise. action space is (1) take a step in current direction, (2) turn right (3) turn left.

Wall perturbation, Fig. 4(b). We randomly sample a point $x_0$ along the x-axis (excluding the first and last columns), and add a vertical wall spanning positions $(x_0, 1)$ through $(x_0, 6)$.

Random walls, Fig. 4(c). Walls are added to 7 blocks randomly sampled from the grid, excluding the start and goal positions.

Goal shift, Fig. 4(d). The goal position is randomly shifted to any block in the grid, excluding the starting position.

---

[5]Available at https://github.com/kingdy2002/VCSE

### B.3 Pusher experiment

The code for our experiments is based on the CleanRL codebase [20], which is released under the MIT License [6] We use PPO with a recurrent network for all baselines. Note that it is known that to maximize state entropy over finite number of rollouts, a non-Markovian policy is necessary. State entropy is calculated over the puck's x,y positions.

**Transition kernel perturbation (wall):** The obstacle is composed of three axis-aligned blocks with width 3cm, centered at (0.52, -0.24), (0.55, -0.21) and (0.58, -0.18). These positions are roughly along the perpendicular bisector of the line between the puck's initial position and the goal. We used 200 episodes of length 100 for evaluating each method. As in [13], to decrease the variance in the results, we fixed the initial state of each episode:

$$qpos = [0., 0., 0., 0., 0., 0., 0., -0.3, 0.2, 0., 0.]$$
$$qvel = [0., 0., 0., 0., 0., 0., 0., 0., 0., 0., 0.]$$

**Action robust baseline [6]** We ran as a baseline both PRMDP and NRMDP. To ensure a fair comparison we implemented the following modifications:

- Larger networks: We increased the width of all neural networks to 256 units.
- hyperparameter sweep: we swept for exploration noise $\in [0.2, 0.6, 1]$, learning rate $\in [1e-3, 5e-4, 1e-4]$ and batch size $\in [256, 512, 1024]$
- Check-pointing: we saved both the best performing checkpoint during training (evaluated in the nominal environment) and the last iteration checkpoint. We evaluated both of them in the perturbed environment and showed the better of the two.

**Reward shift:** The original goal was located at (0.66, -0.35). In the reward shift experiment, a new goal is randomly sampled uniformly from the circumference of a circle with radius r centered at the original goal location. For each radius in [0.1,0.15] we randomly sample 200 goals, evaluate once per goal, and report the average performance across all goals.

**Number of rollouts ablation, Fig. 5(b).** To ensure a fair comparison, we fix the total batch size to 32 across all variants. For a variant using $x$ rollouts per entropy estimate, we divide the 32 rollouts into $\frac{32}{x}$ non-overlapping groups of size $x$, and compute the intrinsic reward separately for each group. As the number of rollouts affects the scale of the intrinsic reward, we tune the temperature individually for each variant.

### B.4 Ant experiment

We train All policies on Mujoco's "ant-v5". To emphasize the affects of each regularization we augment the environment to be less reward shaped in the following way: The episode does not truncate when the ant's in an unhealthy position. Furthermore, there is no reward for being in a healthy position and the penalties on enduring contact forces and control cost are turned off. The reward in our setup is only the forward progress reward. State entropy is calculated over the height and orientations of the ant, encouraging the agent to learn to walk in diverse ways.

**Transition kernel perturbation (vertical wall)** During evaluation, we add an unobservable wall of a size of $[1, \infty, 0.3]$; the only way to pass the wall is to hurdle above it. Evaluation protocol is the same as in the pusher experiment.

### B.5 Interplay between State and Policy Entropy Regularization

In the experiments reported in the main text, we include policy entropy regularization alongside state entropy to ensure stable training, following common practice. Here, we empirically assess whether policy entropy contributes to robustness beyond its stabilizing role.

As shown in Figure 8, omitting policy entropy leads to unstable training, reflected in high variance in nominal performance. To isolate the effect of state entropy, we take a policy trained with both regularizations and continue training it for 2 million steps using only state entropy. As shown in

---

[6]https://github.com/vwxyzjn/cleanrl

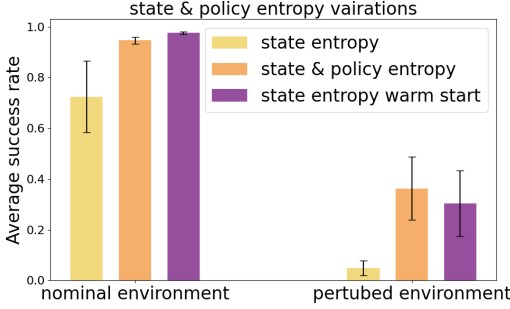

Figure 8: State entropy regularization unstable training and warm start experiment

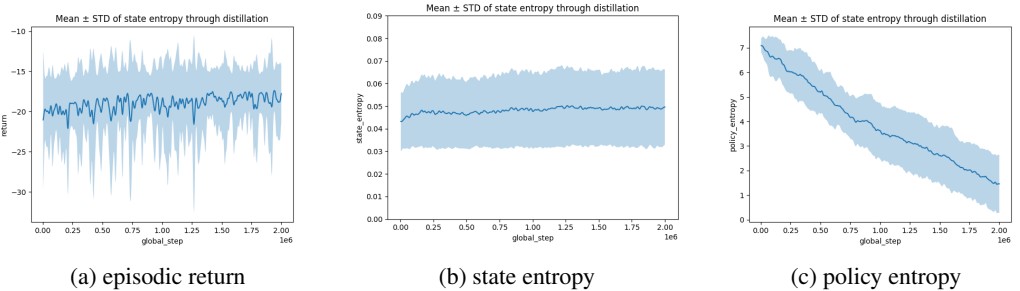

(a) episodic return            (b) state entropy            (c) policy entropy

Figure 9: State and policy entropy -> only state entropy distillation

Figure 9, the policy maintains both its return and state entropy, despite a significant drop in policy entropy. The resulting policy's robust performance closely matches that of the state-and-policy entropy variant.

We hypothesize that with more careful algorithmic tuning, state entropy alone could suffice for both robustness and stability. However, we leave a systematic investigation of its role in training stability to future work.

## B.6 Technical details

All experiments were conducted on a machine equipped with an NVIDIA RTX 4090 GPU. Recreating experiments B.2 takes about 12 hours to run 25 seeds for the 3 baselines, sweeping regularization temperature took an additional day. Recreating experiments B.3 takes a day for 25 seeds for the 3 baselines. rollouts, temperature ablations and regularization temperature searching were another 2 days. Other experiments which didn't make it to the final version took about a week of GPU hours.

