# OpenReview forum: "State Entropy Regularization for Robust Reinforcement Learning"
_NeurIPS.cc/2025/Conference — NeurIPS 2025 oral_

### Official Review · Reviewer_2pKs · 2025-07-01

**Clarity:** 3
**Significance:** 4
**Originality:** 3
**Rating:** 5
**Confidence:** 4

**Summary:**

This paper contrasts State Entropy Regularization (SER) against the more common Policy Entropy Regularization (PER). The claim given is that SER is robust to more global perturbations whereas PER protects against a locally informed adversary. Theoretical results are provided to support these claims. This is followed by a discussion of the limitations of SER, namely that it can hurt performance in risk-averse settings, and is also sensitive to the number of rollouts used to compute the state entropy and evaluate the policy. Empirical results are provided on a discrete and a continuous RL task.

**Questions:**

1. Given your analysis and results, can you provide a structured way to decide on when to use PER, SER, and SEAR? Would SER suffice in every case if we have sufficient compute and parallelization to generate the number of rollouts required for it to work?
2. How scalable will SER be for problems with high-dimensional state-space?

**Ethical Concerns:**

["NO or VERY MINOR ethics concerns only"]

**Final Justification:**

The authors have addressed all of my questions, I have kept my score, and I still recommend acceptance.

**Limitations:**

The authors adequately address the limitations.

**Paper Formatting Concerns:**

There are no concerns

**Quality:**

4

**Strengths And Weaknesses:**

## Strengths

1. The theoretical results comparing SER, PER, and State-Action Entropy Regularization (SEAR) is really valuable from a practical standpoint because depending on the exploration behavior and robustness properties of the regularization, it may or may not be suitable for a particular task.
2. The paper is mathematically rigorous and the notation is consistent and generally easy to follow (see below for some exceptions and suggestions).
3. Given the more theoretical focus of this paper, the empirical results are adequate and substantiate the claims made by the authors (see below for some exceptions and suggestions).

## Weaknesses

1. The authors claim SER is more robust to global perturbations and PER to local. This needs to be made more precise, perhaps with more numerical results. For example, in Table 1, from the column on worst case reward, intuition is provided on why PER tackles adversaries with local information and why SER tackles adversaries with global state distribution information. However, in the example of the MiniGrid environment showing kernel robustness, a single obstruction in the optimal path is considered as a global change, whereas obstructions spread out in state space is considered local. Perhaps a paragraph making it precise what it means to be local and global in this context would be helpful.
2. The derivations in the Appendix skip over quite a bit of steps, which makes it hard to follow in some parts. For example, how do you get to Equation 14 in Appendix A.1, given that (negative) entropy is the convex conjugate of LSE? Given that the space in the Appendix is not limited, giving more explanation on the jumps between different equations will enhance readability quite a bit.

---

> ### Author Rebuttal · Authors · 2025-07-31
>
> Thank you very much for the careful and constructive review. We will make sure to reflect the reviewer's suggestions and clarifications in the revision.
>
> ## Questions
>
> **1. Guidelines for Choosing Between PER, SER, and SEAR**
>
> We thank the reviewer for this interesting and practical question.
>
> These regularizers have additional benefits beyond robustness (e.g., stability and exploration), but these are outside the scope of this paper. From a robustness perspective, the choice of regularization should mainly depend on prior belief about the deployment environment: policy entropy (PER) suits better to small, uniform perturbation, as it spreads stochasticity along a dominant trajectory, whereas for larger, spatially correlated disruptions, state entropy (SER) promotes broader coverage across multiple well\-performing trajectories.
>
> In the absence of clear prior knowledge, our results suggest that, when properly tuned, SER does not harm nominal performance and does not diminish the robustness benefits of PER, so it can be incorporated.
>
> **2. Scalability of SER in High-Dimensional State-Space**
>
> Thank you for raising this important remark. As discussed in the paper e.g., when using the k\-NN estimate, state entropy estimation indeed becomes more statistically demanding as the dimensionality of the state space increases. In practice, this can be mitigated by computing entropy over a subspace of relevant state features selected based on prior knowledge of the test environment, or by estimating entropy over latent state representations, as demonstrated in prior work [1,2,3,4,5]. These approaches help scaling SER to high-dimensional problems.
>
> ## Weaknesses
>
> **1. Clarifying the Notion of Local vs. Global Perturbations**
>
> We appreciate the reviewer’s comment and recognize that the distinction between “local” and “global” could benefit from more consistent wording, particularly when using these terms to describe types of perturbations and behaviors. We will keep this in mind and use clearer, more precise language in the revision.
>
> A more accurate way to phrase our results is that SER improves robustness to spatially correlated perturbations—or equivalently, against adversaries that exploit global properties of the agent’s behavior across the state space. In contrast, PER primarily improves robustness against perturbations that are not strongly state-specific or spatially structured.
>
> For example, in MiniGrid, the wall-type perturbation places obstacles in highly correlated locations, creating a major local blockage. In contrast, in the spread\-out perturbation, each cell is obstructed independently with a small uniform probability, leading to more scattered, uncorrelated disruptions.
>
> Each regularizer induces a behavior that aligns with the type of robustness it provides: to handle spatially correlated perturbations, an SER\-agent learns to take more diverse trajectories, exhibiting more “global” behavior, as expected when maximizing state entropy. Conversely, a PER agent spreads randomness only along a single dominant trajectory, exhibiting more “local” behavior that yields robustness primarily against smaller, more uniform perturbations.
>
> **2. Improving Clarity and Detail in Appendix Derivations**
>
> We thank the reviewer for the helpful suggestion. We will expand the appendix derivations by adding the missing intermediate steps.
>
> Specifically, for the proof in Appendix A.1 \- equation 13 implies that $\min_{\tilde r \in \tilde R}\mathbb{E}\_{\rho^\pi}[\tilde r] = \mathbb{E}\_{\rho^\pi}[r]-\epsilon + \min_{\Delta r} (-\langle d^\pi,\Delta r^\pi \rangle +\text{LSE}_{\mathcal{S}}(\Delta r^\pi))$. The fact that negative entropy is the convex conjugate of LSE implies that $\min\_{\Delta r} (-\langle d^\pi,\Delta r^\pi \rangle +\text{LSE}\_{\mathcal{S}} (\Delta r^\pi)) = \mathcal{H}\_{\mathcal{S}} (d^\pi)$. By substituting this in we get our desired result $\min\_{\tilde r \in \tilde R} \mathbb{E}\_{\rho^\pi}[\tilde r] = \mathbb{E}\_{\rho^\pi}[r]-\epsilon + \mathcal{H}\_{\mathcal{S}} (d^\pi)$.
>
> ---
>
> References:
>
> [1] Hao Liu and Pieter Abbeel. Behavior from the void: Unsupervised active pre-training. In Advances in Neural Information Processing Systems, 2021.
>
> [2] Younggyo Seo, Lili Chen, Jinwoo Shin, Honglak Lee, Pieter Abbeel, and Kimin Lee. State entropy maximization with random encoders for efficient exploration. In International Conference on Machine Learning, 2021.
>
> [3] Dongyoung Kim, Jinwoo Shin, Pieter Abbeel, and Younggyo Seo. Accelerating reinforcement
> learning with value-conditional state entropy exploration. In Advances in Neural Information
> Processing Systems, 2023.
>
> [4] Dongyoung Kim, Jinwoo Shin, Pieter Abbeel, Younggyo Seo. Accelerating Reinforcement Learning with Value-Conditional State Entropy Exploration. In Advances in Neural Information Processing Systems (NeurIPS) 2023
>
> [5] Denis Yarats, Rob Fergus, Alessandro Lazaric, Lerrel Pinto. Reinforcement Learning with Prototypical Representations. In International Conference on Machine Learning, 2021.

---

> ### Comment · Reviewer_2pKs · 2025-08-01
>
> Thank you for the reply, I have no further questions, and would keep my score as it is.

---

### Official Review · Reviewer_6Gze · 2025-07-02

**Clarity:** 3
**Significance:** 3
**Originality:** 3
**Rating:** 5
**Confidence:** 3

**Summary:**

+ Summary & Contributions
	- This paper offers a mixture of theoretical and empirical contributions concerning the use of state entropy regularization in RL.
	- Various theoretical results are provided to elucidate properties of state entropy regularization and draw explicit connections between classic RL with state entropy regularization and standard robust RL within an uncertainty set over the reward function.
	- Insights into the practical aspects of employing state entropy regularization are also provided with regards to the suitable sample size needed to obtain reliable entropy estimates.
	- Empirical results are provided across both MiniGrid and Mujoco environments illustrating that, while nominal performance is comparable and minimally affected by the presence of state entropy regularization, there can be statistically significant improvements in policy robustness. Ablations are included to assess the impact of sample size and the free temperature parameter used in characterizing the reward uncertainty sets.

**Questions:**

I believe my questions for the authors should be apparent in my review above.

**Ethical Concerns:**

["NO or VERY MINOR ethics concerns only"]

**Final Justification:**

During the rebuttal period, the authors addressed my major concerns regarding quality and significance. They have also productively engaged with and resolved concerns raised by other reviewers. With that, I maintain my original score for acceptance.

**Limitations:**

Yes

**Quality:**

3

**Strengths And Weaknesses:**

+ Quality
	- Strengths
		- Aside from the comment below concerning Theorem 4.1, all the technical proofs seem to be in order.
		- The main theorems in Section 3 do a good job of walking the reader through the novel connection between state entropy regularization and robust RL with reward uncertainty sets.
	- Weaknesses
		* Major
			- Theorem 4.1 seems rather strange. It would be one thing to assert that there exists a MDP where a certain property does or does not hold. It seems like quite another thing to assert that there does not exist any MDP where a property holds by illustrating that the property does not hold for a single, specific MDP. Either I've misunderstood something about the goals of Section 4.1 or Theorem 4.1 is nowhere near as general as the authors claim it is. The verbatim of Theorem 4.2 (that is, a "there exists a MDP" claim) seems more in line with what is shown in the proof.
		* Minor
			- In Appendix A.3, there is no such thing as "l’hospital’s rule" but there is a result known as L'Hôpital's rule.
			- The CVaR result, while clearly important towards risk-averse RL, seems like a bit of an "odd man out" relative to the other results. It would have been nice to see an empirical result on risk-averse RL to make CVaR seem less like an after thought.

+ Clarity
	- Strengths
		- The paper is well-written and reasonably structured.
	- Weaknesses
		* Major
			- N/A
		* Minor
			- N/A


+ Originality
	- Strengths
		- While I am not intimately familiar with the existing literature on robust RL, the provided theoretical results are novel, to the best of my knowledge, while appropriately extending existing results (credited by citation) as needed.
	- Weaknesses
		* Major
			- N/A
		* Minor
			- N/A

+ Significance
	- Strengths
		- Improving the robustness of deep RL algorithms continues to be an important area of research. While the scale of the empirical results reported in this work is modest, this seems perfectly acceptable for a paper leaning much more heavily on clean theoretical analysis and insights.
		- Empirical results demonstrate clear robustness benefits of state entropy regularization and the experiments have been conducted with an ample number of seeds to draw meaningful conclusions.
	- Weaknesses
		* Major
			- As a RL researcher who does not spend much time thinking about robust RL, I can only truly confirm that the theoretical results provided appear to be correct without having much insight into their broader significance and impact on this sub-area of the RL literature. Furthermore, I don't get the impression that any one of the theorems proven in this paper constitute a significant standalone advance; however, it may be the case that the collective insight of the theorems is the valuable theoretical contribution upon which this paper rests. I would be curious to hear the authors' thoughts on the impact of their theoretical contributions and how they perceive the advancement stemming from their contributions to existing work on robust RL.
		* Minor
			- N/A


+ Final Remarks
	- Overall, I think this paper scores nicely on all axes and the issues I've raised can be resolved/clarified quite quickly from the authors via the rebuttal process. My main hestiation is on the perceived impact of this work for the robust RL literature, which is already a difficult and subjective thing to forecast and measure in any paper. With that in mind, I'll recommend acceptance for the paper and look forward to hearing from the authors during the rebuttal process on the few points I've raised.

---

> ### Author Rebuttal · Authors · 2025-07-31
>
> We thank the reviewer for the thoughtful and encouraging review. We appreciate the time and care dedicated to evaluating our work, and will make sure to incorporate all suggestions and clarifications into the last version.
>
> ## Weaknesses
>
> **1. Quality: Clarifying the Scope of Theorem 4.1**
>
> Thank you for raising this point, we understand how the scope of Theorem 4.1 may have been unclear. To clarify, we say that a kernel-uncertainty problem defined by $\tilde{\mathcal{P}}(\mathrm{MDP}, \pi)$ is equivalent to a regularized objective defined by $\Omega(\mathrm{MDP}, \pi)$ if for every MDP and policy $\pi$ it holds that $\min\_{\tilde{P} \in \tilde{\mathcal{P}}}\mathbb{E}\_{\rho\_\tilde{P}^\pi}[r] = \mathbb{E}\_{\rho\_P^\pi}[r] + \Omega$. This notion of equivalence is the same as the one used earlier in the paper when showing the relationship between reward robustness and entropy regularization, and aligns with definitions in prior robustness literature (e.g., [1]). Here, we focus on problems where both the uncertainty set $\tilde{\mathcal{P}}$ and the regularization $\Omega$ depend only on the nominal kernel $P$ and the policy $\pi$. In this setting, exhibiting a single MDP where the equality fails for all $\tilde{\mathcal{P}}$ and $\Omega$ is sufficient to disprove universal equivalence.
>
> The intended takeaway of Theorem 4.1 is not that equivalence fails for all MDPs, but rather that for every uncertainty which is a function only of $P$ and $\pi$, there does not exist a regularizer—also depending only on $P$ and $\pi$—that can represent the robust return for all MDPs. We will rephrase the theorem in the revision to better reflect this intended level of generality.
>
> **2. Quality: Minor Comments**
>
> Thank you for pointing this out—we will correct the spelling to L’Hôpital’s rule.
>
> Regarding the CVaR result, we included it as a theoretical complement to highlight an important limitation of state entropy in risk-averse settings. We agree that it may feel out of the main paper's thread and will reduce its emphasis in the revision to keep the narrative more focused (e.g., remove the preliminary mention in the main text).
>
> **3. Significance: Theoretical Contributions**
>
> While we do provide novel individual results, we agree that a primary contribution lies in the collective insight of our analysis.
>
> For robust RL, we show state entropy regularization induces robustness to spatially correlated perturbations—a class of perturbations that is highly relevant in settings like transfer learning, yet remains sparsely addressed in the robust RL literature.
>
> Additionally, maximum state entropy is well-studied for exploration and coverage, but never explicitly intended for robust RL. Here we highlight the robustness benefits and practical considerations associated with this regularizer, which we believe sheds new light on this important line of work. Our analysis clarifies the type of robustness it implicitly provides and emphasizes key practical aspects—namely, its sensitivity to rollout budgets for reliably realizing these robustness benefits.
>
> ---
>
> References:
>
> [1] Benjamin Eysenbach and Sergey Levine. Maximum entropy RL (provably) solves some robust RL problems. In International Conference on Learning Representations, 2022.

---

> ### Comment · Reviewer_6Gze · 2025-08-06
>
> I thank the authors for their time and effort in providing a rebuttal response to my review.
>
> I appreciate their clarification regarding Theorem 4.1 and now understand the nature of their claim. I would concur that some amount of rewording in the claim and proof would go along way to clarifying the result for other readers.
>
> I also thank the authors for their candid framing around the significance of their contributions, and was glad to hear that we are in alignment regarding the holistic nature of the contributions.
>
> It seems there is one other reviewer who scored the paper negatively. I will inspect their review and your rebuttal response to them independently. For now, my concerns have been addressed and I see no reason to adjust my score.

---

### Official Review · Reviewer_i4aV · 2025-07-03

**Clarity:** 4
**Significance:** 3
**Originality:** 3
**Rating:** 5
**Confidence:** 4

**Summary:**

The paper investigates the theoretical guarantees and practical implications of using state entropy regularization in Reinforcement Learning (RL) to improve robustness. While state entropy regularization has recently been empirically shown to outperform policy entropy measures (i.e. action conditional entropy), its theoretical underpinnings regarding robustness have not received much attention.

The main contributions are:
- The paper characterizes robustness succinctly, following prior work, stating that state entropy regularization enhances robustness against spatially correlated perturbations (i.e. a local adversary).
- The analysis contrasts state entropy with the widely used policy entropy regularization. It shows that policy entropy regularization protects only against local adversaries, whereas state entropy provides robustness against globally informed perturbations (by encouraging state space coverage).
- The paper derives theoretical guarantees, largely following prior work.

**Questions:**

Please discuss the weaknesses outlined above. In particular:
- It would be good for me to confirm my sense of how the authors think their work fits into the landscape of theoretical results and how practically relevant the derived results actually are.
- It would be good to understand if the authors have had the chance to experiment with less toy-ish domains.
- The "fully kernel robust" setting doesn't strike me as very practical (and isn't achievable as mentioned in the paper). This isn't a strong weakness in the analysis but I wonder if there are more practical robustness settings one should worry about and analyse?

**Ethical Concerns:**

["NO or VERY MINOR ethics concerns only"]

**Final Justification:**

The authors addressed my concerns adequately and thus I am raising my score to an accept.

**Limitations:**

The paper does a good job at not only selling one method but highlighting limitations throughout and discussing them.

**Quality:**

3

**Strengths And Weaknesses:**

Strengths:
- I worked my way through the theoretical derivations once and while I am not an expert RL theory the derivations seem sound (albeit somewhat close to prior work, see comment below).
- (Policy) entropy regularization is highly used in current SOTA RL algorithms and a paper characterizing it further and explaining differences between different regularization schemes is relevant to the community.
- The paper includes some insightful findings that I would not have guessed before reading it. E.g. the paper illustrates that the robustness benefits of state entropy are more sensitive to the number of rollouts used for policy evaluation compared to policy entropy. This sensitivity is attributed to the statistical difficulty of accurately estimating state entropy and behavioral aspects in low-rollout regimes. Another surprising find (at least without thinking carefully about it) is that state and action entropy regularization together can be worse than state entropy regularization by itself in some cases.
- The paper is generally well written and easy to follow despite it's heavily theoretical content.
- The presented didactic (toy) examples (e.g. Fig.1 and Fig.3) do a good job at explaining the issues with policy entropy regularization and the sensitivity of state entropy regularization.
- I like that the paper doesn't try to sell their analysis or experiments of one method but rather tries to give a balanced view of strengths and weaknesses.

Weaknesses:
- The derivation of the theoretical results seems like a relatively straightforward extension to known results for policy entropy regularization (which is already known to solve a robust RL problem). The relevant works are cited and the extensions cover a case that's clearly relevant to the community. Nonetheless it does dampen the strict "novelty" of this part a bit.
- To the best of my understanding the analysis largely covers the discrete/tabular case and it's applicability to more practical settings is somewhat limited. The authors also acknowledge that in practice some positive effects of policy entropy regularization are observed that aren't fully explainable by the theory.
- Minor there is some notation that is a bit loose in the main paper (e.g. Dissimilarity of reward functions isn't formally defined, definition of adversary is a bit vague etc.) I think this is ok as it aids the flow of the paper not having too much notation, but I could imagine some readers with a more theoretical background searching for some definitions.
- State entropy is notoriously hard to estimate and work with (this is also somewhat covered by the theoretical results re. sensitivity to rollouts). So perhaps a strong guarantee on the case where it can be estimated nearly exactly is not as impactful in practice.
- The experiments are on fairly toyish domains. The MiniGrid example clearly is more didactic than practically relevant. The control domain (pusher) is good but the only reasonably high dimensional domain considered. It would have been nice to see results on some standard domains or some more practically relevant domains.

---

> ### Author Rebuttal · Authors · 2025-07-31
>
> We thank the reviewer for the thoughtful and constructive feedback. Below, we respond to the main points raised below, and will incorporate relevant clarifications and improvements into the revised version.
>
> ## Questions
>
> **1. Clarifying the Positioning and Significance of our Analysis**
>
> We thank the reviewer for the question and appreciate the opportunity to clarify.
>
> While previous studies have focused on the connection between robustness and policy entropy, our work is the first to thoroughly characterize the robustness properties of state entropy regularization—including its benefits, limitations, and interaction with policy entropy—both theoretically and empirically. In particular, we show that state entropy naturally induces robustness to spatially correlated perturbations, a class of real-world deviations (e.g., in transfer learning) that remains underexplored in the robust RL literature—not only in the context of entropy regularization.
>
> Moreover, maximum state entropy is commonly used for practical purposes such as exploration and coverage [1,2,3,4,5], not explicitly for robustness. By analyzing the type of robustness it provides and its practical challenges—especially its reliance on rollout budgets—we aim to inform both the theoretical understanding and empirical use of this regularizer.
>
> **2. Additional Experiments**
>
> We indeed had the opportunity to experiment with additional domains beyond those presented in the main paper and plan to incorporate these results in the revision.
>
> - We first ran an additional experiment on Ant in MuJoCo, where the goal is to progress as far as possible along the $y$-axis within an episode. At test time, we introduce a hurdle perpendicular to this axis. To continue progressing, the agent must jump over the obstacle. As shown in the table below, the agent trained with state entropy regularization significantly outperforms the unregularized agent and the one regularized with policy entropy.
>
>
> | agent\performance | nominal env -reward | perturbed env -obstacle pass rate |
> | -------- | -------- | -------- |
> | Unregularized | 802.65+-97.96|0.13+-0.04 |
> | Policy entropy| **827.25+-85.72**|0.21+-0.07|
> | State entropy | 795.01+-122.32|**0.36+-0.07** |
>
> - We also ran an experiment illustrating reward robustness in continuous control. In this experiment conducted in Pusher, we randomized the puck's target location at test time by uniformly sampling it within a circle around the nominal goal. As shown in the table below, the agent trained with state entropy regularization achieves better performance under this variation compared to other baselines.
>
> | agent\reward shift radius | no shift | 0.1 |0.15|
> | -------- | -------- | -------- | -------- |
> | Unregularized | **0.98+-0.02**|0.53+-0.05 | 0.30+-0.03   |
> | Policy entropy| 0.94+-0.017|0.65+-0.041|0.31+-0.04     |
> | State entropy | 0.95+-0.011|**0.71+-0.039**  |**0.35+-0.03**   |
>
>
> - We finally extended the original Pusher experiment by adding action-robust baselines, which seem to be standard in robust RL for continuous control [6]. As shown in the table below, state entropy regularization significantly outperforms these baselines. Nonetheless, we must emphasize that we do not expect state entropy to consistently outperform robust RL methods across all types of perturbation. However, this result—along with our accompanying analysis—suggests that it can be particularly effective in scenarios involving spatially correlated perturbations, something that standard methods are not designed to address.
>
> | agent\task | nominal env |perturbed env - wall|
> | -------- | -------- | -------- |
> | Unregularized | 0.98+-0.02|0.005+-0.01|
> | Policy entropy| 0.94+-0.017|0.03+-0.03|
> | State entropy | 0.95+-0.011|**0.37+-0.12** |
> | PRMDP[6] | 0.98+-0.015|0.02+-0.03 |
> | NRMDP[6]| **1+-0**|0.03+-0.06 |
>
>
> **3. On the Practicality of Kernel-Robust RL**
>
> Thank you for raising this point. Generalizing RL agents to environments that differ from training is a fundamental challenge with many important applications (e.g., sim-to-real transfer). While multiple formulations and approaches address this challenge (e.g., [9]), one prominent and well-studied framework is robust RL, which explicitly models uncertainty and optimizes for the worst-case performance (e.g., [10, 6, 11]). Within this framework, kernel robustness remains a widely used and practical formulation—particularly in settings where test-time shifts are hard to anticipate and performance must remain reliable under a range of plausible deviations.
>
> We also clarify that our negative result regarding kernel-robust MDPs does not imply that this setting is intractable in practice. Rather, it shows that no regularizer depending only on the policy and transition kernel can capture the robust return across all MDPs. In practice, effective solutions exist for practically useful uncertainty sets (e.g., [6] presents an effective solution for a rectangular uncertainty set).
>
> ## Weaknesses
>
> **1. On the Novelty of the Results**
>
> We acknowledge that our analysis builds on similar tools used in prior work on policy entropy and robust RL—a connection the reviewer rightly notes is explicitly acknowledged in the main paper. That said, while we adopt similar techniques, the results themselves are novel. Furthermore, we believe the insights and perspectives developed throughout the paper—summarized concisely in our response to Question 1—are themselves original and offer meaningful value to the community.
>
> **2. On Tabular/Discrete Assumptions**
>
> We thank the reviewer for the thoughtful comment. Analyzing the tabular case is a standard approach in the literature, as it allows us to characterize robustness in a tractable setting while still offering insights applicable to more complex domains. We also note that most of the results can be extended to the continuous case with relatively minor adjustments, and we will add a note on this in the revision.
>
> **3. Minor Comments**
>
> We thank the reviewer for the suggestion and will revise the text to clarify key terms while maintaining readability.
>
> **4. State Entropy for Robustness Under Noisy Estimation**
>
> The reviewer is making a great point that the estimation error of the entropy may degrade the robustness guarantees in practice. We can quantify the degradation by specializing the results of Th. 3.1 (reward robustness) and Th. 3.3 (kernel uncertainity) for the estimated entropy regime. Let $\hat d$ be the empirical estimate of $d^\pi_P$ obtained with $n$ sampled trajectories. We can prove that
> $$
> \mathcal{H} (d^\pi_P) \geq \mathcal{H} (\hat d) - \sqrt{\frac{2S \log (2 / \delta)}{n}} \log \sqrt{\frac{S n}{\log (2 /\delta)}}
> $$
> holds with probability $1 - \delta$ for $\delta \in (0, 1)$. Then, we can plug the right-hand side in place of $\mathcal{H} (d^\pi_P)$ into Eq. 6 and 7 to obtain sample-based high porbability lower bounds on the robustness. The sample-based bounds approximate the exact entropy results at a $O(n^{-1/2})$. We will add these results to the paper together with their proof, which is based on centration of the empirical disitribution [7] and continuity bounds of the entropy [8].
>
> **5. Regarding Experimental Domains**
>
> We thank the reviewer for the comment. Please see our response to Question 2, where we present additional experiments we plan to include in the revised version.
>
> ---
>
> References:
>
> [1] Hao Liu and Pieter Abbeel. Behavior from the void: Unsupervised active pre-training. In Advances in Neural Information Processing Systems, 2021.
>
> [2] Younggyo Seo, Lili Chen, Jinwoo Shin, Honglak Lee, Pieter Abbeel, and Kimin Lee. State entropy maximization with random encoders for efficient exploration. In International Conference on Machine Learning, 2021.
>
> [3] Dongyoung Kim, Jinwoo Shin, Pieter Abbeel, and Younggyo Seo. Accelerating reinforcement
> learning with value-conditional state entropy exploration. In Advances in Neural Information
> Processing Systems, 2023.
>
> [4] Dongyoung Kim, Jinwoo Shin, Pieter Abbeel, Younggyo Seo. Accelerating Reinforcement Learning with Value-Conditional State Entropy Exploration. In Advances in Neural Information Processing Systems (NeurIPS) 2023
>
> [5] Denis Yarats, Rob Fergus, Alessandro Lazaric, Lerrel Pinto. Reinforcement Learning with Prototypical Representations. In International Conference on Machine Learning, 2021.
>
> [6] Chen Tessler, Yonathan Efroni, and Shie Mannor. Action robust reinforcement learning and applications in continuous control. In International Conference on Machine Learning, pp. 6215–6224. PMLR, 2019.
>
> [7] Tsachy Weissman, Erik Ordentlich, Gadiel Seroussi, Sergio Verdu, Marcelo J. Weinberger. Inequalities for the L1 Deviation of the Empirical Distribution. 2003.
> [8] Andreas Winter. Tight uniform continuity bounds for quantum entropies. 2015.
>
> [9] Robert Kirk, Amy Zhang, Edward Grefenstette, and Tim Rocktäschel. 2023. A Survey of Zero-shot Generalisation in Deep Reinforcement Learning. J. Artif. Int. Res. 76 (May 2023).
>
> [10] Lerrel Pinto, James Davidson, Rahul Sukthankar, and Abhinav Gupta. Robust adversarial reinforcement learning. In International Conference on Machine Learning, pages 2817–2826. PMLR, 2017.
>
> [11] Uri Gadot, Kaixin Wang, Navdeep Kumar, Kfir Y. Levy, and Shie Mannor. 2024. Bring your own (non-robust) algorithm to solve robust MDPs by estimating the worst kernel. In Proceedings of the 41st International Conference on Machine Learning (ICML'24), Vol. 235. JMLR.org, Article 576, 14408–14432.

---

> > ### Comment · Reviewer_i4aV · 2025-08-07
> >
> > Thank you for the additional experiments and explanations.
> >
> > While they naturally cannot entirely address the concern regarding the theory deviating from practice they do strengthen the paper and the additional experiments seem well thought outout. I would hope that the authors find a nice way to include them in the final version of the paper.
> > Given that my concerns are largely addressed I have raised my score.

---

> > > ### Author Response · Authors · 2025-08-07
> > > **Thank You for the Follow-Up**
> > >
> > > Thank you for your thoughtful follow-up and for taking the time to re-engage with the discussion. We appreciate your updated assessment and will incorporate the additional material into the final version of the paper.

---

### Official Review · Reviewer_rqBU · 2025-07-03

**Clarity:** 3
**Significance:** 2
**Originality:** 2
**Rating:** 4
**Confidence:** 4

**Summary:**

The paper studies the use of state entropy regularization in robust RL. The problem setup is about robust RL where the environment parameters such as the transition kernel and the reward function are not fully known. In such situations, robust RL agents are trained to adapt to any slight changes in these quantities. The authors discuss why policy entropy regularization is suboptimal. They derive uncertain reward sets for state entropy regularization and also show that state-entropy regularization can produce a lower bound for transition kernel uncertainty as well. The authors test their regularization on Minigrid and Mujoco Pusher environments.

**Questions:**

(1) Is there a better way to estimate the visitation probabilities/entropy?

(2) How would you actually maximize reward + visitation entropy? As this reward now is non-markovian (policy dependent)?

**Ethical Concerns:**

["NO or VERY MINOR ethics concerns only"]

**Final Justification:**

The authors clarified my major concerns.

**Limitations:**

The major limitation is the applicability. The authors experiment on very simple domains with a specialized way of estimating state-visitation distribution which questions where this method can be applied. A thorough experimental evaluation with several baselines is also missing.

**Paper Formatting Concerns:**

nil

**Quality:**

3

**Strengths And Weaknesses:**

Strength

(1) Changes in reward functions at test time is a very practical problem so coming up with a simple robust RL method that is mathematically motivated makes sense.

(2) The authors derive the reward uncertainty sets for state-entropy regularized RL.

(3) The authors show that state-entropy regularized RL produce a lower bound for transition kernel uncertainty.

(4) The authors also show that using entropy based regularizations alone, transition kernel based uncertainty cannot be completely solved.

Weakness

(1) Discussions on the physical examples of the reward uncertainty set is missing. There should be some discussion about physically what changes in the reward function are allowed.

(2) The method for computing state-visitation distribution (using k-nearest neighbours) is limiting.

(3) Experiments are also limiting. No real baseline is used apart from policy entropy regularization. Are there other robust RL algorithms?

(4) Missing references:

[1]: Lee, L., Eysenbach, B., Parisotto, E., Xing, E., Levine, S., & Salakhutdinov, R. (2019). Efficient exploration via state marginal matching. arXiv preprint arXiv:1906.05274.

[2]: Agarwal, S., Durugkar, I., Stone, P., & Zhang, A. (2023). f-policy gradients: A general framework for goal-conditioned rl using f-divergences. Advances in Neural Information Processing Systems, 36, 12100-12123.

---

> ### Author Rebuttal · Authors · 2025-07-31
>
> Thank you for taking the time to carefully review our paper and provide constructive feedback. We appreciate the reviewer's comments and will incorporate them into the revised version.
>
> ## Questions
>
> Before answering the reviewer's questions below, we emphasize that these aspects have been thoroughly studied in prior works on state entropy maximization [e.g., 1, 2, 3, 4, 5, 6, 7] and fall outside the scope of our work.
>
> **1. Is there a better way to estimate the visitation probabilities/entropy?**
>
> Possibly, but this is not the focus of our work. In the RL setting, k-NN entropy estimation [1, 2, 3, 4, 5] outperforms alternative methods such as computing the entropy from KDE [6] or learned density models such as VAEs (e.g., [7] mentioned by the reviewer). It has become a standard approach for state entropy maximization, thus justifying our choice.
>
> **2. How would you actually maximize reward + visitation entropy? As this reward now is non-markovian (policy dependent)?**
>
> To maximize the reward + visitation entropy, we follow state-of-the-art approaches from [1, 3, 4, 5] by combining the external reward with an "entropy"-reward derived from the k-NN estimator (implementation details are reported in Appx. B.1). We understand the reviewer's concern: the entropy-reward is technically non-Markovian. However, [1, 3, 4, 5] have shown that it can practically be optimized using standard RL techniques.
>
> ## Limitations
>
> **1. Applicability as a major limitation**
>
> The method used for optimizing the state entropy regularization as analyzed in this paper has been successfully applied in challenging tasks, including high-dimensional continuous control [2] and visual domains [1, 3, 5].
> Therefore, we do not think applicability is a concern and we focus our study on other aspects (robustification induced by the regularization).
> In principle, our findings should apply to any domain in which state entropy maximization has been successfully deployed.
>
> ## Weaknesses
>
> **1. Physical examples of the reward uncertainty set**
>
> Thank you for raising this point. Since the reviewer's comment may reflect two distinct concerns, we address both interpretations below—one regarding the interpretation of the uncertainty set, the other regarding reward uncertainty experiments on a physical domain.
>
>  **1.1. Bridging Theory and Intuition for the Reward Uncertainty Set**
>
> We acknowledge that the uncertainty set may initially seem abstract and we will clarify its intuition in the revised version. Specifically, as suggested, we will add a worked example on a simple two-state, two-action MDP illustrating that the method is robust against an adversary with a soft exponential budget (the paper already includes a basic illustrative example in Figure 1). In addition, we believe that highlighting how the uncertainty set interpolates between more familiar $\ell_\infty$-like (localized, uniformly bounded) and $\ell_1$-like (global, budgeted) perturbations can provide further interpretability. Finally, while the set itself couples rewards through a log-sum-exp form, we would like to emphasize that the worst-case perturbations remain intuitive: policy entropy allows only local statewise attacks, whereas state entropy enables global attacks concentrated on frequently visited states.
>
> **1.2. Empirical Illustration of Reward Uncertainty in Pusher**
>
> If the reviewer’s comment refers to the absence of physical examples involving reward perturbations, we have prepared an additional experiment in the Pusher domain that we will include in the revised version. In this experiment, we simulate reward uncertainty by randomizing the puck’s goal location at test time, uniformly sampling from a circle around the nominal target. This setup captures a natural class of perturbations arising from slight spatial shifts in task objectives. As shown in the table below, the agent trained with state entropy regularization achieves better performance under this variation compared to other baselines.
>
> | agent\reward shift radius | no shift | 0.1 |0.15|
> | -------- | -------- | -------- | -------- |
> | Unregularized | **0.98+-0.02**|0.53+-0.05 | 0.30+-0.03   |
> | Policy entropy| 0.94+-0.017|0.65+-0.041|0.31+-0.04     |
> | State entropy | 0.95+-0.011|**0.71+-0.039**  |**0.35+-0.03**   |
>
>
> **2. Regarding the Use of k-NN State-Occupancy Entropy Estimation**
>
> Please see our answer to Question 1.
>
>
> **3. Additional Robust Baselines and Domains**
>
> We thank the reviewer for raising this important point.
>
> First, we extended the original Pusher experiment by adding a standard robust RL baseline [8]. As shown in the table below, the agent trained with state entropy regularization significantly outperforms this baseline. While we do not expect state entropy to consistently outperform robust RL methods across all types of perturbations, this result—along with our analysis in the main paper—suggests that it is particularly effective against spatially correlated perturbations, which are not the primary focus of most standard approaches.
>
> | agent\task | nominal env |perturbed env - wall|
> | -------- | -------- | -------- |
> | Unregularized | 0.98+-0.02|0.005+-0.01|
> | Policy entropy| 0.94+-0.017|0.03+-0.03|
> | State entropy | 0.95+-0.011|**0.37+-0.12** |
> | PRMDP[8] | 0.98+-0.015|0.02+-0.03 |
> | NRMDP[8]| **1+-0**|0.03+-0.06 |
>
> Second, we added an additional experiment on the standard Ant benchmark in MuJoCo. In this setting, the agent is trained to progress along the $y$-axis. At test time, we introduce a hurdle perpendicular to this axis that the agent must jump over. As shown in the table below, the agent trained with state entropy regularization significantly outperforms both the unregularized agent and the one regularized with only policy entropy.
>
> | agent\performance | nominal env -reward | perturbed env -obstacle pass rate |
> | -------- | -------- | -------- |
> | Unregularized | 802.65+-97.96|0.13+-0.04 |
> | Policy entropy| **827.25+-85.72**|0.21+-0.07|
> | State entropy | 795.01+-122.32|**0.36+-0.07** |
>
> **4. Missing References**
>
> We thank the reviewer for pointing out these relevant works. We will include both in the revised related work section.
>
> ---
>
>
> **References:**
>
> [1] Hao Liu and Pieter Abbeel. Behavior from the void: Unsupervised active pre-training. In Advances in Neural Information Processing Systems, 2021.
>
> [2] Mirco Mutti, Lorenzo Pratissoli, and Marcello Restelli. Task-agnostic exploration via policy gradient of a non-parametric state entropy estimate. In AAAI Conference on Artificial Intelligence, 2021.
>
> [3] Younggyo Seo, Lili Chen, Jinwoo Shin, Honglak Lee, Pieter Abbeel, and Kimin Lee. State entropy maximization with random encoders for efficient exploration. In International Conference on Machine Learning, 2021.
>
> [4] Dongyoung Kim, Jinwoo Shin, Pieter Abbeel, Younggyo Seo. Accelerating Reinforcement Learning with Value-Conditional State Entropy Exploration. In Advances in Neural Information Processing Systems (NeurIPS) 2023
>
> [5] Denis Yarats, Rob Fergus, Alessandro Lazaric, Lerrel Pinto. Reinforcement Learning with Prototypical Representations. In International Conference on Machine Learning, 2021.
>
> [6] Elad Hazan, Sham Kakade, Karan Singh, Abby van Soest. Provably efficient maximum entropy exploration. In International Conference on Machine Learning 2019.
>
> [7] Lee, L., Eysenbach, B., Parisotto, E., Xing, E., Levine, S., & Salakhutdinov, R. (2019). Efficient exploration via state marginal matching. arXiv preprint arXiv:1906.05274.
>
> [8] Chen Tessler, Yonathan Efroni, and Shie Mannor. Action robust reinforcement learning and applications in
> continuous control. In International Conference on Machine Learning, pp. 6215–6224. PMLR, 2019.

---

### Note · Authors · 2025-08-15

Dear AC and Reviewers,

Thank you for your constructive feedback and for overseeing the discussion. We provide below a brief summary of the rebuttal phase for your convenience.

To the best of our understanding, after the rebuttal phase all reviewers have a positive outlook on our paper (minimum score ≥ 4), with most agreeing on accept (average 4.75). Specifically, rqbu and i4av raised their scores, while 6gze and 2pks kept their initial high ratings (5).

After the rebuttal, the reviewers seem to agree that the paper provides the first thorough characterization of the robustness properties of state-entropy regularization—including its benefits, limitations, and interaction with policy entropy—supported by both theoretical analysis and experiments. For the robust RL community, this matters because the paper shows that state-entropy regularization naturally induces robustness to spatially correlated perturbations—a practically relevant yet underexplored class of deviations (e.g., in transfer learning). Furthermore, these insights are also important because maximum state entropy is used for exploration and coverage rather than explicitly for robustness; our analysis clarifies the specific form of robustness it implicitly provides, and the practical considerations—most notably the sensitivity to rollout budgets—needed to reliably realize these benefits.

While reviewers raised some concerns, our understanding from the discussion is that these were all addressed through clarifications and additional results; as noted above, all reviewers either raised their score or kept their already high one.

We appreciate the valuable discussion and are ready to integrate all additions and clarifications in the final version.

Sincerely,
The authors

---

### Decision · Program_Chairs · 2025-09-17

**Decision:**

Accept (oral)

**Comment:**

(a) Scientific Claims and Findings

The paper's central scientific contribution is the formal characterization of state entropy regularization's effect on policy robustness. The key finding is that maximizing the entropy of the state visitation distribution is equivalent to solving a robust RL problem against a "globally informed" adversary that can introduce structured, spatially correlated perturbations to the reward function. This is contrasted with policy entropy, which is shown to protect against "locally informed" adversaries that make small, state-specific changes. The paper also provides a performance lower bound under transition kernel uncertainty and formally proves the limitations of the approach, namely that entropy regularization alone cannot solve general kernel-robust problems and can be detrimental in risk-averse scenarios.

(b) Strengths

The paper has several strengths. First, it addresses a clear and important gap in the literature, providing the first formal analysis of a widely used technique (state entropy maximization) through the lens of robustness. Second, the analysis is rigorous and comprehensive, presenting a balanced view that includes not only positive results (duality theorems, performance bounds) but also important limitations (impossibility results, risk-averse degradation, sample sensitivity). Third, the paper is very clear and well-written, making complex theoretical concepts accessible through intuition and illustrative examples. Finally, the empirical results, especially after being augmented during the rebuttal, provide convincing support for the theoretical claims across different domains and perturbation types.

(c) Weaknesses

The initial submission had some weaknesses, primarily concerning the limited scope of the empirical validation and the lack of comparison to standard robust RL baselines. Please note that the theoretical techniques are mainly trivial extensions of prior work, though applied in a novel context and well explained. However, these weaknesses were almost entirely addressed by the authors' thorough rebuttal, which included new experiments on more challenging tasks (Ant) and against relevant robust RL methods, thereby substantiating their claims more forcefully.

(d) Reasons for Decision

The recommendation to accept is based on the paper's significant contribution, technical quality, and potential impact. The paper provides novel and valuable insights into the connection between exploration and robustness, clarifying what kind of robustness state entropy provides. This is of high practical importance, as it offers guidance on when this form of regularization is most appropriate.

(e) Discussion and Rebuttal

The initial concerns of the reviewers were thoughtfully and comprehensively addressed with new results that substantially strengthened the paper, leading to a unanimous and enthusiastic consensus for acceptance. Strong recommendation to include the new experiments and the clarification on the theory in the final version